# Elongator mutation in mice induces neurodegeneration and ataxia-like behavior

Marija Kojic 1, Monika Gaik[2], Bence Kiska[1], Anna Salerno-Kochan[2,3], Sarah Hunt[4], Angelo Tedoldi 4, Sergey Mureev[1], Alun Jones[1], Belinda Whittle[5], Laura A. Genovesi[1], Christelle Adolphe[1], Darren L. Brown[1], Jennifer L. Stow 1, Kirill Alexandrov[1], Pankaj Sah[4], Sebastian Glatt 2 & Brandon J. Wainwright[1]

Cerebellar ataxias are severe neurodegenerative disorders with an early onset and progressive and inexorable course of the disease. Here, we report a single point mutation in the gene encoding Elongator complex subunit 6 causing Purkinje neuron degeneration and an ataxia-like phenotype in the mutant *wobbly* mouse. This mutation destabilizes the complex and compromises its function in translation regulation, leading to protein misfolding, proteotoxic stress, and eventual neuronal death. In addition, we show that substantial microgliosis is triggered by the NLRP3 inflammasome pathway in the cerebellum and that blocking NLRP3 function in vivo significantly delays neuronal degeneration and the onset of ataxia in mutant animals. Our data provide a mechanistic insight into the pathophysiology of a cerebellar ataxia caused by an Elongator mutation, substantiating the increasing body of evidence that alterations of this complex are broadly implicated in the onset of a number of diverse neurological disorders.

[1] Institute for Molecular Bioscience, the University of Queensland, Brisbane, QLD 4072, Australia. [2] Max Planck Research Group at the Malopolska Centre of Biotechnology, Jagiellonian University, Krakow, Poland. [3] Postgraduate School of Molecular Medicine, Warsaw, Poland. [4] Queensland Brain Institute, University of Queensland, Brisbane, QLD 4072, Australia. [5] John Curtin School of Medical Research, Australian National University, Canberra, Australia. Correspondence and requests for materials should be addressed to S.G. (email: sebastian.glatt@uj.edu.pl) or to B.J.W. (email: b.wainwright@imb.uq.edu.au)

A taxias are the most common neurological deficit resulting from cerebellar dysfunction[1,2]. These are progressive and currently incurable disorders with gradual deterioration in signs and symptoms, most commonly due to neurodegeneration of an unknown etiology. Affected neurons are Purkinje neurons (PNs) and, rarely, granule neurons. Previous mouse models of cerebellar ataxia have provided insight into the neuropathology of the disease[3], however, the precise molecular mechanisms of neuronal loss remain largely unknown.

Using an ab initio N-ethyl-N-nitrosourea (ENU) mutagenesis screen, here we identify a yet uncharacterized *Elp6* mutation, which perturbs the stability and function of the murine Elongator complex, and results in a severe ataxic phenotype, the *wobbly* mouse, with associated microgliosis and degeneration of cerebellar PNs. Elp6 is one of the six subunits (Elp1–6) of the highly conserved eukaryotic Elongator complex, which is organized in two subcomplexes, namely Elp123 and Elp456. It has been shown that all Elongator subunits equally contribute to the stability, integrity, and functionality of the complex in yeast[4]. Involvement of the complex in various cellular processes, including transcription, cell motility, cytoskeleton organization, exocytosis and intracellular trafficking, has been highlighted by a number of reports[5]. However, recent studies provide evidence that the above mentioned functions assigned to the complex are downstream effects of its master activity as a global translational regulator[6,7]. In detail, Elongator-dependent modifications of uridines in the wobble position of tRNA anticodons seem to be of key importance for the fidelity and kinetics of translational elongation, which also guides and directs cotranslational folding dynamics[8,9]. Over the past decade, a number of studies have showed that the Elongator complex is involved in various cellular activities that govern the development and maintenance of the nervous system[10–14]. Moreover, several studies have linked the occurrence of specific mutations in Elongator subunits with the onset of various neurological disorders[15–20]. Our results define a mechanism in the pathology of cerebellar ataxias whereby subtle deregulation of tRNA function caused by the *Elp6* mutation, leads to protein misfolding, proteome aggregation, and consecutive neuronal death resulting in a severe manifestation of the disease.

Microgliosis following neuronal loss is a normal physiological response to injury, but when this usually transient event becomes chronic and self-propagating, it can lead to sustained neurodegeneration[21,22]. Hence, microglia do not just provide neuroprotection, but can also promote neurotoxicity. Inflammasomes play a central role in microglia activation, being multiprotein complexes that sense various cellular and environmental stress signals[23]. The NLRP3 inflammasome is expressed and functional in brain microglia[24], and associated with neurodegenerative disorders such as Alzheimer's disease[25], Parkinson's disease[26], multiple sclerosis[27], and prion-like diseases[28], and is the only inflammasome known to be activated by misfolded proteins and their aggregates via a yet not fully defined mechanism[29]. It has been suggested that the activation of the NLRP3 inflammasome occurs in response to infection or injury, and involves consequential NLRP3 oligomerization, which serves as a scaffold to nucleate an apoptosis-associated speck-like protein containing a caspase recruitment domain (ASC) that further acts as a docking platform for pro-caspase-1. Ultimately, NLRP3 activation leads to activation of caspase-1, which itself promotes pro-interleukin-1β (IL-1β) processing and release of the mature cytokine IL-1β[30]. Subsequently, cytokine release promotes inflammation and leads to further damage to neurons already primed for degeneration. Here, we demonstrate that blocking microglial priming by inhibiting the NLRP3 pathway

can attenuate PN degeneration and the ataxic phenotype in *wobbly* mice.

## Results

**Cerebellar ataxia and PN degeneration in *wobbly* mice.** The *wobbly* mouse was identified in a recessive ENU-mutagenesis screen on the basis of its wobbly gait. Ataxic symptoms commence at postnatal day (P) 60 in the form of loss of gait coordination and balance, reduced locomotor activity, and abnormal hindlimb clasping, which is commonly observed in mice with a neurodegenerative defect[31] (Fig. 1a and Supplementary Fig. 1). The phenotype gradually becomes more pronounced and is most severely manifested by P100. The mutant mice also showed an impaired performance on the rotarod, balance beam and tests in the Catwalk system (Supplementary Fig. 2). The defect was found to occur in a recessive manner, as the phenotype of heterozygous mice was indistinguishable from that of the wild-type animals (Fig. 1a and Supplementary Figs. 1 and 2). Notably, no differences were observed in the performance of male versus female *wobbly* animals in any of the performed tests (Supplementary Figs. 1 and 2). The mice show no alternations in overall life span, weight, and fertility.

Histological analyses at P60 revealed a specific and substantial loss of PNs in *wobbly* mice, as we were not able to detect any other changes in overall cerebellar morphology (Fig. 1b). Neuropathology was restricted to PNs in the cerebellum with other central nervous system structures and nonneuronal tissue being unaffected (Supplementary Fig. 3). The first histological signs of PN degeneration were already detectable at P40 and the cerebellum of mutant mice was completely depleted of PNs by P120 (Fig. 1c). Consistent with previous findings that Zebrin II can act as a neuroprotector[32,33], we found that PNs of *wobbly* mice that did not express this molecular marker were more susceptible to cell death than Zebrin II-immuno-positive cells (Supplementary Fig. 4a). Individual variations in the severity of the *wobbly* phenotype manifestation appeared due to different magnitudes rather than localization of PN loss, which was shown to be consistent across all cerebellar functional subdivisions, from anterior to posterior lobe, across vermis and hemispheres (Supplementary Fig. 4b–j).

To test the functional impact of an underlying mutation, electrophysiological recordings were obtained from PNs in *wobbly* mice at P21–P24, before motor symptoms were apparent. Already at this stage, PNs exhibited altered passive and active membrane properties (Fig. 1d–f and Supplementary Table 1), and an altered synaptic excitation/inhibition balance (Supplementary Fig. 5 and Supplementary Table 1). In summary, mutant PNs were less excitable showing increased resting membrane potential and action potential threshold and being able to generate significantly fewer action potentials. Parallel fiber stimulation failed to evoke excitatory synaptic currents (EPSCs) in more than 50% of PNs, while evoked synaptic inhibition was shown to be stronger.

***Elp6L126Q* underlies *wobbly* mouse phenotype.** Whole-exome sequencing of *wobbly* mice identified a T/A substitution in the *Elp6* gene (Fig. 2a), leading to a single amino acid L/Q substitution at position 126 in the protein (*Elp6L126Q*). To further define Elp6 function in vivo, we generated Elp6 knockout (KO) mice (Supplementary Fig. 6a). Consistent with previous studies that have identified developmental defects following Elp1 and Elp3 ablation[20,34], we found that a loss of Elp6 (*Elp6−/−*) also results in early embryonic lethality (Supplementary Fig. 6b, c). To confirm that the *Elp6L126Q* allele drives the pathology observed in the *wobbly* mutant, we crossed *Elp6+/−* to *wobbly* mice and screened

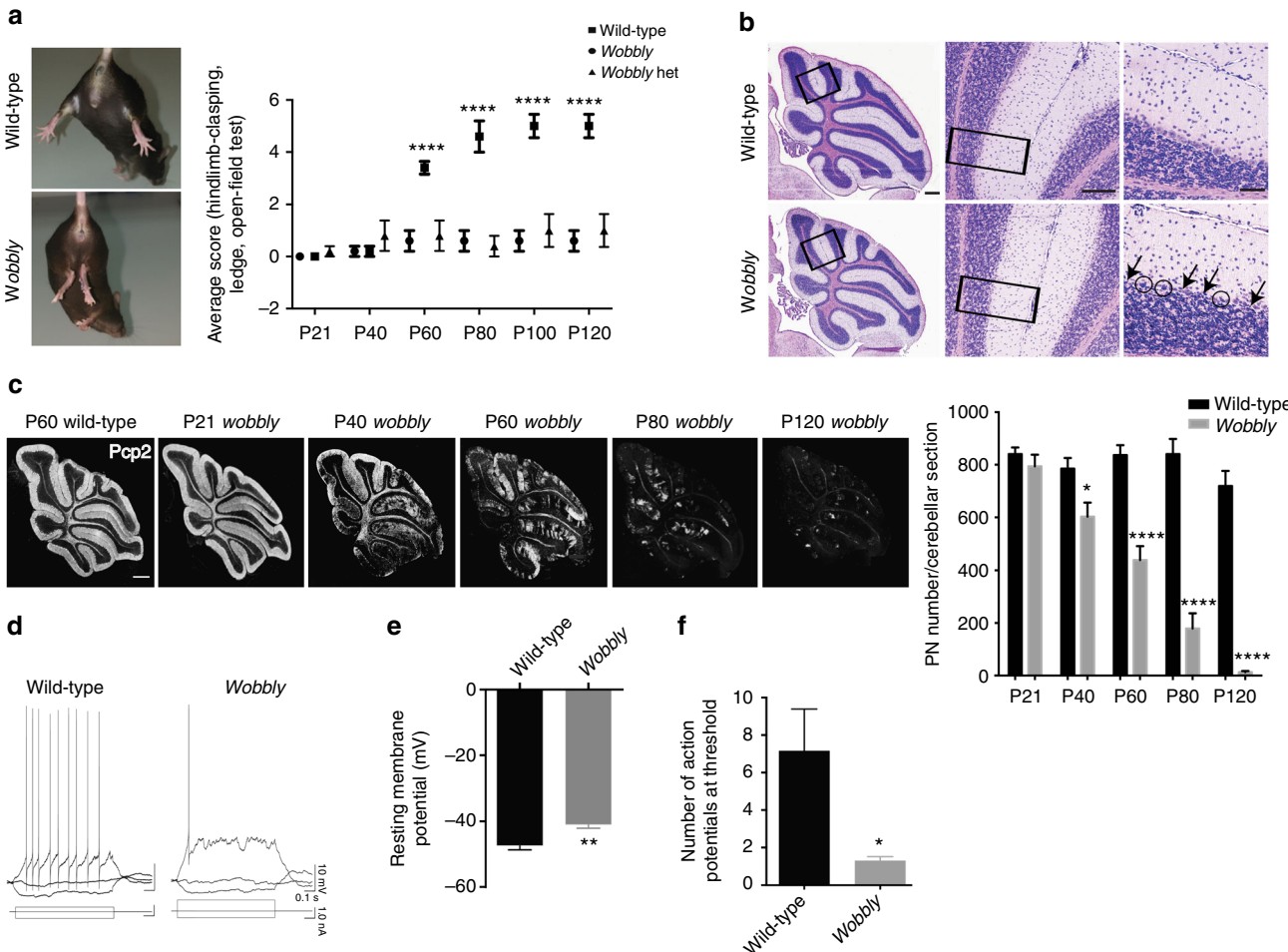

**Fig. 1** Ataxia and neurodegeneration in *wobbly* mice. **a** Left: hindlimb clasping in wild-type and mutant animals. Right: behavioral analysis of the *wobbly* phenotype relative to wild-type (*n* = 10 (5 males and 5 females) for each of the genotypes; homozygous *wobbly* animals are presented as *wobbly* and heterozygous as *wobbly* het). **b** H&E staining of P60 cerebellar sagittal. Arrows indicate PN loss, and circles mark degenerating PNs. Black rectangles represent magnified areas. **c** Pcp2 immunofluorescence and PN quantification in *wobbly* and wild-type cerebella. **d** Representative raw traces of train of action potentials in PNs elicited by current injections of 50 and 100 pA in wild-type and *wobbly* cells, respectively. **e** Resting membrane potential of *wobbly* (−40.9 ± 1.2 mV, *n* = 10) and wild-type (−47.3 ± 1.4 mV, *n* = 18) PNs at P21–24. **f** Number of action potentials of *wobbly* (*n* = 10) and control (*n* = 13) PNs at threshold elicited by current injections. For (**b**) and (**c**) *n* = 5 for each of the genotypes and ages presented; representative images are shown. Scale bars: (**b**) left panel, (**c**) 500 µm; (**b**) middle panel, 100 µm; (**b**) right panel, 50 µm. Statistical evaluation: (**a**, **c**) two-way ANOVA and Sidak's multiple comparisons test; (**e**, **f**) two-tailed *t* test. Statistically significant differences are indicated (*$P \leq 0.05$; **$P \leq 0.01$; ****$P \leq 0.0001$). Data represent mean ± SEM

their progeny for main features of the phenotype. The compound heterozygous (*Elp6*[L126Q/−]) animals expressed a more rapidly progressive phenotype than *wobbly* mice, demonstrating that *Elp6L126Q* likely functions as a hypomorph. Furthermore, *Elp6*[L126Q/−] displayed a decreased survival rate, with 80% of animals not surviving beyond P40 (Fig. 2b). Histological and behavioral analyses of these mice demonstrated a full recapitulation of the *wobbly* phenotype (Fig. 2c, d). No phenotypic abnormalities were found in *Elp6*[+/−] animals, indicating that one copy of wild-type *Elp6* is sufficient for proper functioning of the Elongator complex.

In agreement with its here described role, in situ hybridization confirmed wide expression of the Elp456 subcomplex throughout the cerebellum (Supplementary Fig. 7). To establish whether *Elp6L126Q*-mediated degeneration was an intrinsic defect of PN neurons, a transgenic rescue of the *wobbly* phenotype was initiated by crossing *wobbly* mice to transgenic animals expressing the HA-tagged wild-type *Elp6* driven by the PN-specific *Pcp2* promoter (*Pcp2-Elp6-HA*; Fig. 3a). Ataxic phenotype and neurodegeneration in *Pcp2-Elp6-HA*; *wobbly* mice were

fully rescued by the transgene (Fig. 3b–d), demonstrating that despite the widespread expression of the Elongator complex in the cerebellum, PN degeneration is mainly initiated cell-autonomously.

**Elp6L126Q negatively affects stability and function of the Elongator complex.** Taking advantage of the high sequence conservation of the Elongator subunits among eukaryotes[35], the recently determined crystal structure of the yeast Elp456 (yElp456) subcomplex[36], and the electron microscopy reconstruction of the fully assembled Elongator complex[37], we were able to assess the precise location of the mutated residue within the Elongator complex (Fig. 4a). Despite its detrimental effects, *Elp6L126Q* resides on the periphery of the complex, distant from known tRNA-, ATP-, SAM-, or acetyl-CoA binding sites and on the side of the Elp456 ring, which is located opposite from the enzymatically active Elp3 subunit and proposed tRNA binding and modification pocket. To understand the consequences of the *Elp6L126Q* mutation on the molecular level, we produced

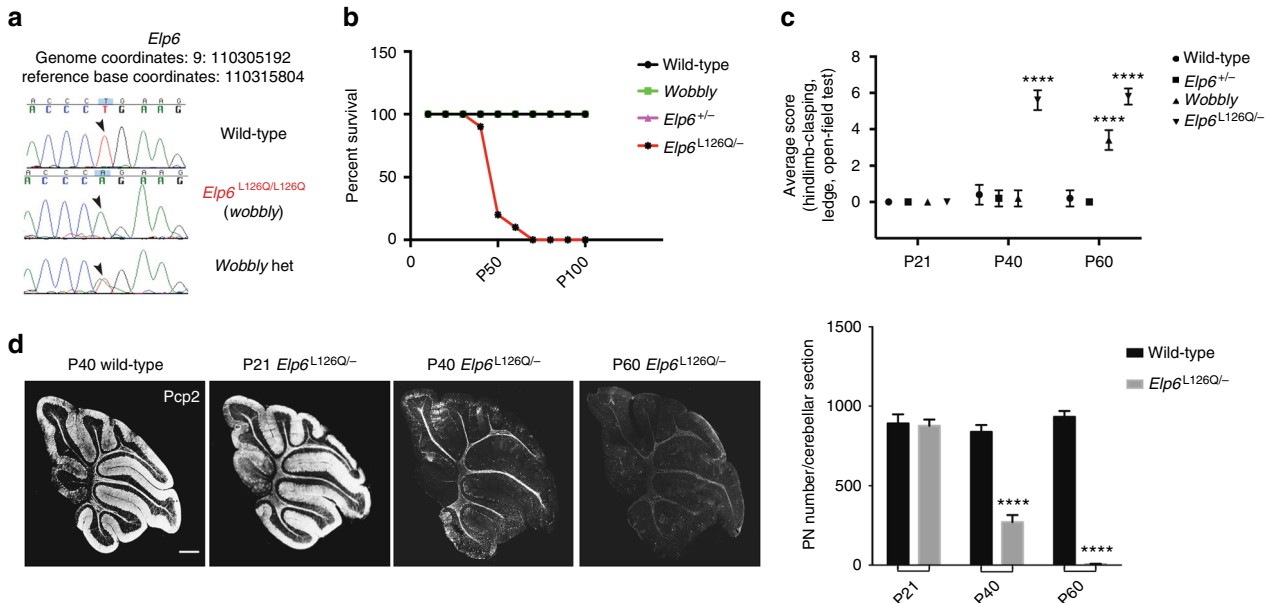

**Fig. 2** The *wobbly* mutation is identified in the *Elp6* gene. **a** Sequencing chromatograms showing the *Elp6L126Q* mutation. Arrowheads indicate the affected nucleotide. **b** Survival rates of wild-type, *wobbly*, *Elp6*[+/−] and *Elp6*[L126Q/−] mice ($n = 30$ animals for each of the genotypes). **c** Ataxic phenotype scoring of *wobbly* and *Elp6*[L126Q/−] mice relative to wild-type ($n = 6$ (3 males and 3 females) for each of the genotypes). **d** Pcp2 immuno-labeling and quantification of PNs in *Elp6*[L126Q/−] and control animals ($n = 5$ for each of the genotypes and ages presented; representative images are shown). Scale bar: 500 μm. Statistical evaluation: two-way ANOVA and Sidak's multiple comparisons test. Statistically significant differences are indicated ([****]$P ≤ 0.0001$). Data represent mean ± SEM

recombinant mElp456 in bacteria and purified it to homogeneity. The three murine subunits, like yElp456, are able to form a dimeric Elp56 intermediate and a hexameric Elp456 assembly (Fig. 4b and Supplementary Fig. 8a). The introduced *Elp6L126Q* mutation leads to the destabilization of mElp6 itself (Supplementary Fig. 8b) and the disappearance of the Elp56 intermediate, but still permits hexamer formation (Fig. 4b and Supplementary Fig. 8c). Notably, we observed decreased thermal stability of *Elp6L126Q*-containing mElp456 hexamers in comparison to wild-type mElp456 (Fig. 4c and Supplementary Fig. 8d). We detected the same effect for full length and truncated versions of Elp456 (lacking predicted disordered regions in the N- and C-termini of Elp4 and Elp5), indicating that the mutation indeed affects the integrity of the core of the complex. The decrease in protein stability appeared to be coupled to reduced tRNA binding capacity in vitro (Supplementary Fig. 8e) and reduced levels of wobble base modification (ncm⁵U and mcm⁵U) in brains of *wobbly* mice in vivo (Fig. 4d). As the decreased in vitro tRNA binding capacity of mutated mElp456 is not restricted to Elongator-modifiable tRNAs, it most likely results from the decreased stability of the hexameric assembly and does not affect the tRNA selection mechanisms. The induced appearance of aggregates in the *Elp6L126Q* sample at elevated temperatures and nanoDSF analyses independently confirm our thermofluor data and support the notion of a complex stability related reduction in tRNA affinity (Supplementary Fig. 8f). Other tRNA modifications, which are not conducted by Elongator, did not show decreased levels in the analyzed samples. This data demonstrate that *Elp6L126Q* destabilizes the complex and perturbs its function in the regulation of translation.

**Protein misfolding and aggregation in *wobbly* PNs**. Given that perturbation of wobble uridine modifications was shown to lead to ribosome pausing and protein misfolding and aggregation[8], we next performed ultrastructural analyses of degenerating neurons to screen for signs of proteotoxic stress. The analyses using transmission electron microscopy revealed extensive autophagy,

apoptotic cell features, and an accumulation of electron-dense globular structures that likely represent protein inclusions (Fig. 5a). No signs of necrosis and excitotoxicity were found in *wobbly* mouse PNs. Increased ubiquitination and upregulation of the molecular chaperone Hsp70 observed in *wobbly* PNs confirmed the presence of protein aggregates and induced proteotoxic stress response (Fig. 5b). Detection of activated caspase-3 supported apoptosis-mediated cell death. In addition, we show that these cells express the endoplasmic reticulum (ER) stress-induced transcription factor CHOP. This further suggests that apoptosis in *wobbly* mouse PNs is likely induced by the ER stress elicited by unfolded protein response, as previously described in various other neurodegenerative diseases[38].

**NLRP3-mediated inflammation contributes to the *wobbly* pathology**. Next, we found prominent histopathological signs for substantial microgliosis (Iba1-marked glial population) (Fig. 6a–c) coupled to reactive astrogliosis (GFAP-marked glial population; Supplementary Fig. 9) in the cerebellum of *wobbly* mice. The appearance of these inflammatory markers is confined to the cerebellum and strictly associated with PN degeneration. Given that several studies in the past decade have demonstrated that neuroinflammation can be initiated by the inflammasome complexes in microglia activated by protein aggregates[23], we further checked for inflammasome activation in our mutant mice cerebella. Notably, the cerebella of the ataxic *wobbly* mice showed a strong upregulation of key inflammasome effectors, including cleaved caspase-1[39] and ASC[40] (Fig. 6b-d). ASC was shown to be overexpressed and formed so-called specks, previously demonstrated to activate cytokine cascade.

Given the accumulation of protein aggregates in *wobbly* PNs and recent findings that the NLRP3 inflammasome can specifically act as a sensor for intracellular misfolded proteins[29], we used a potent and selective NLRP3-inhibitor MCC950[41] to test whether activation of the NLRP3 inflammasome contributes to the progression of neurodegeneration in *wobbly* mice. Treatment of *wobbly* mice with MCC950 significantly delayed

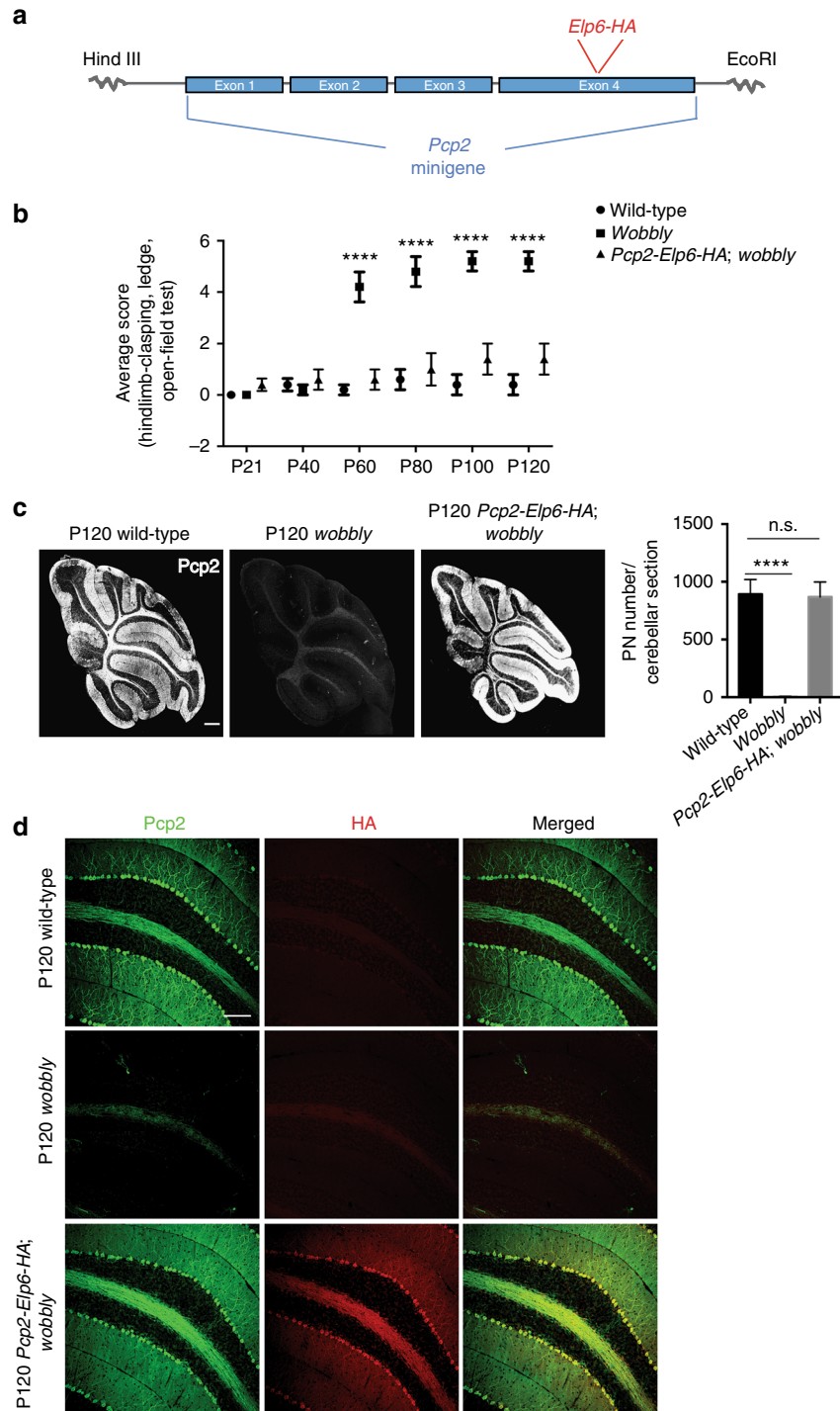

**Fig. 3** Transgenic complementation of the Elp6 defect. **a** Generation of the *Pcp2-Elp6-HA* transgenic construct. **b** Ataxic phenotype scoring of *wobbly* and *Pcp2-Elp6-HA*; *wobbly* mice relative to wild-type ($n = 6$ (3 males and 3 females) for each of the genotypes). **c** PN immuno-staining and quantification in transgenic (*Pcp2-Elp6-HA*; *wobbly*) mice relative to wild-type controls at P120. **d** Immunofluorescent staining of P120 wild-type, *wobbly* and transgenic sagittal sections with anti: Pcp2 or HA antibody. For (**c**) and (**d**) $n = 5$ for each of the genotypes; representative images are shown. Scale bars: (**c**) 500 µm; (**d**) 100 µm. Statistical evaluation: two-way ANOVA and Sidak's multiple comparisons test. Statistically significant differences are indicated (**** $P \leq 0.0001$). Data represent mean ± SEM

the onset of ataxia (Fig. 7a) and decreased the rate of neurodegeneration (Fig. 7b), as a consequence of reduced inflammation and inflammasome activity (Fig. 7c–e). To verify these findings, we took a parallel genetic approach by crossing *wobbly* mice to NLRP3 null animals. Analysis of the ataxic and neurodegenerative features of the double mutant progeny and

controls showed that NLRP3 deficiency in vivo mirrored the phenotype of MCC950-treated mutant mice (Fig. 8a, b), further confirming that the NLRP3-driven inflammatory response contributes to the progressive neuropathology in *wobbly* animals. Reduction of the neuroinflammation in NLRP3 KO; *wobbly* mice was shown to be even stronger than in MCC950-treated mice

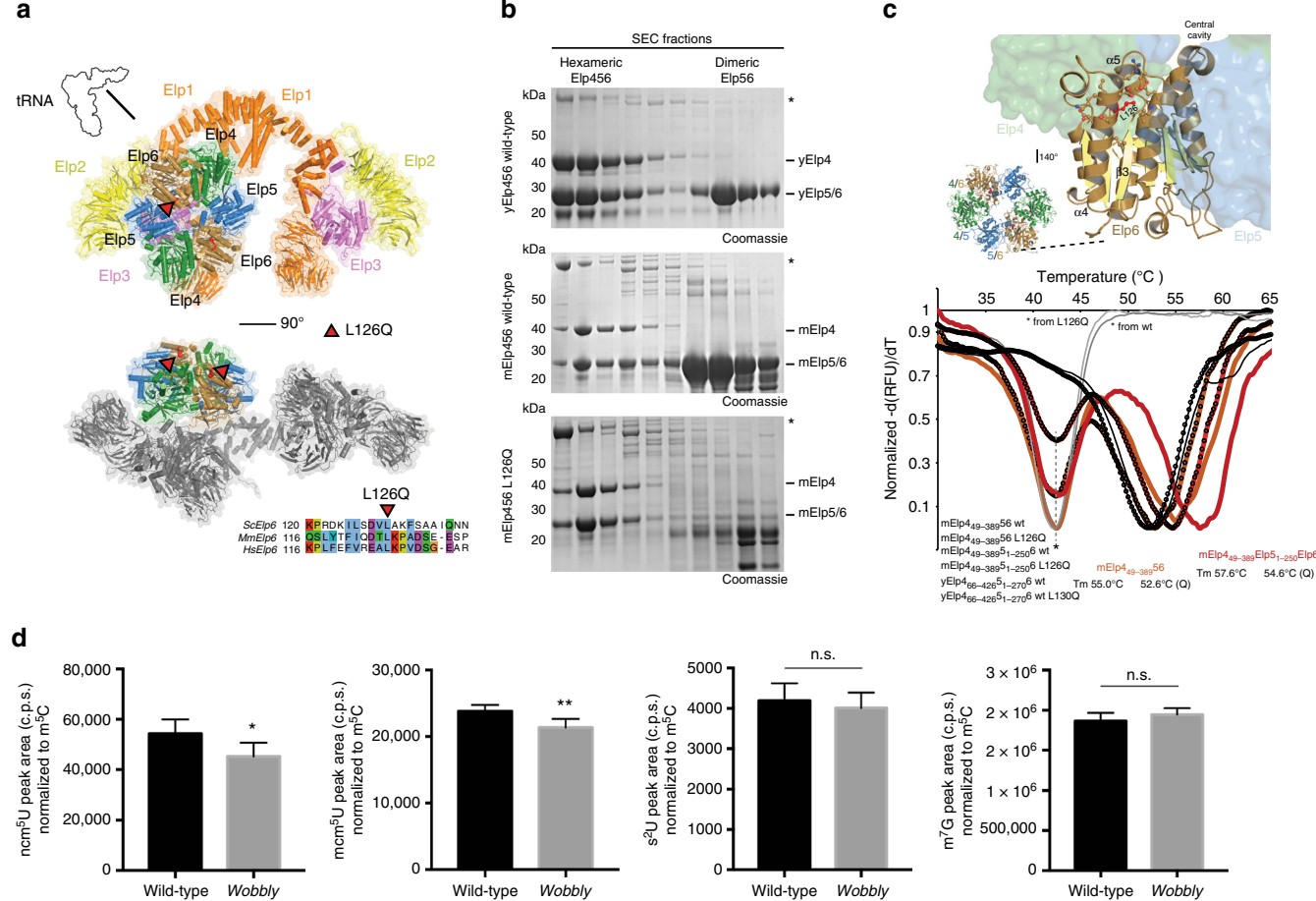

**Fig. 4** *Elp6L126Q* destabilizes Elp456. **a** Structural overview of the Elongator complex in cartoon representation showing individual subunits (Elp1 (orange), Elp2 (yellow), Elp3 (pink), Elp4 (green), Elp5 (blue), and Elp6 (brown)) from two perspectives. The *Elp6L126Q* mutation is highlighted in red (triangles). Multiple sequence alignment of Elp6 proteins from *S. cerevisiae* (Q04868), *M. musculus* (Q8BK75), and *H. sapiens* (AAH00623) highlighting the mutated region (bottom). **b** SDS-PAGE analyses after size exclusion chromatography (SEC) of purified Elp456 complexes from yeast (top), mouse wild-type (middle) and mouse *Elp6L126Q* (bottom). Subunits are labeled on the right and co-migrating chaperone is indicated by an asterisk. **c** same as (**a**) highlighting the localization of the hexameric Elp456 ring and the mutation (top). Normalized first derivative curves from Thermofluor analyses of purified yElp4$_{66-426}$5$_{1-270}$6 (black), mElp4$_{49-389}$56 (orange), and mElp4$_{49-389}$5$_{1-250}$6 (red) complexes. Wild-type complexes are shown by lines, mutations (L126Q for mouse and L130Q for yeast) are labeled with black circles, copurified contaminations are shown in gray. Calculated melting temperatures are indicated. **d** Quantification of ncm$^5$U, mcm$^5$U, s$^2$U, and m$^7$G nucleosides in cerebellar lysates from *wobbly* and wild-type animals analyzed using HPLC/MS (*n* = 5 animals per genotype). Presence of ncm$^5$U and mcm$^5$U tRNA modifications is Elongator-dependent, whilst s$^2$U and m$^7$G are used here as an example of Elongator-independent tRNA modifications. Statistical evaluation: two-tailed *t* test. Statistically significant differences are indicated (*$P \leq 0.05$; **$P \leq 0.01$). Data represent mean ± SEM

(Supplementary Fig. 10a), likely due to a relatively short half-life time of the administered drug[41]. Genetic ablation of caspase-1 function in *wobbly* mice not only reinforced the role of NLRP3 in the progression of neurodegeneration, but also identified caspase-1 as a potential therapeutic target in gliosis-associated ataxias in addition to NLRP3 itself (Fig. 8c, d). Microgliosis and ASC nucleation were found to be reduced to a lesser extent in caspase-1 KO; *wobbly* animals in comparison to NLRP3 KO; *wobbly* and MCC950-tretaed *wobbly* mutants, which is expected given that the inflammatory cascade in these mutants is inhibited downstream of ASC nucleation[30].

## Discussion

Our data indicate a mechanistic route underlying neurodegeneration in cerebellar ataxias based on the perturbed function of the Elongator complex in the regulation of translation as a consequence of the destabilizing *Elp6L126Q* mutation.

The mutation was identified in a mouse model for cerebellar ataxia, the *wobbly* mouse, which develops severe ataxic symptoms as a consequence of extensive Purkinje neurodegeneration. We characterized the ataxic wobbly phenotype in mice and found the induced neurodegeneration on the one hand to follow the pattern of neuroprotective Zebrin II expression and on the other hand to affect all cerebellar functional regions equally. We also showed that intrinsic cellular and synaptic changes of mutant PNs occur prior to clear pathologically recognizable degeneration, which is consistent with observations in other progressive neurodegenerative diseases, such as Alzheimer's disease[42].

On the molecular level, *Elp6L126Q* destabilizes the assembly and integrity of the heterohexameric Elp456 subcomplex, which is ultimately necessary for tRNA binding and tRNA modification activity of the Elongator complex. Furthermore, we demonstrated that this mutation negatively affects the Elongator activity by detecting lower levels of tRNA modifications in cerebella of

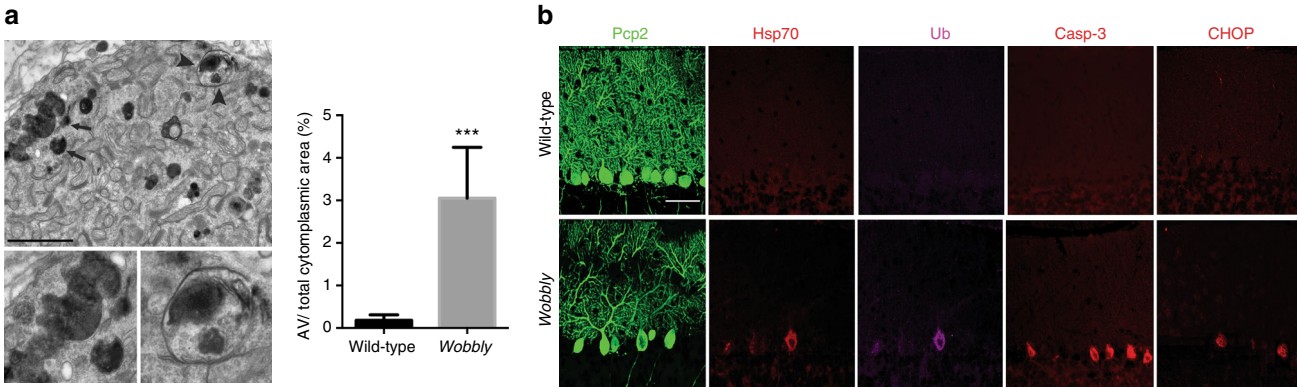

**Fig. 5** Protein misfolding and ER-stress-induced apoptosis in *wobbly* PNs. **a** Electron micrographs and quantification of autophagic vacuoles (AV) in *wobbly* PNs at P40. Arrows indicate electron-dense protein aggregates and arrowheads point to protein aggregates in autophagic bodies, both are magnified in the lower panel (*n* = 3 animals per genotype; *n* = 5 cells per animal; representative images are shown). Statistical evaluation: two-tailed *t* test. Statistically significant difference is indicated (***$P \leq 0.001$). Data represent mean ± SEM. **b** Immunofluorescence with antibodies to Pcp2, Hsp70, ubiquitin (Ub), caspase-3 (Casp-3), and CHOP on P40 wild-type and *wobbly* cerebella (*n* = 5 for each of the genotypes; representative images are shown). Scale bars: (**a**) 2 µm; (**b**) 50 µm

**Fig. 6** Microgliosis and inflammasome activation in *wobbly* mice cerebella. **a** Immunofluorescence of wild-type and *wobbly* cerebella with Pcp2 and Iba1 antibodies. **b** Iba1 and ASC immuno-labeling of *wobbly* and control mice cerebellar sections. Arrows indicate ASC specks in microglia. White rectangles represent magnified areas. **c** Quantification of microgliosis in (**a**) and ASC specks in (**b**). **d** Western blot and quantification of cleaved caspase-1 (Casp-1) and ASC expression in P120 *wobbly* cerebellar brain lysates relative to control. For all experiments *n* = 5 for each of the genotypes; representative images and blots are shown. Scale bars: (**a**) 50 µm; (**b**) 10 µm. Statistical evaluation: (**c**) two-way ANOVA and Sidak's multiple comparisons test; (**d**) two-tailed *t* test. Statistically significant differences are indicated (**$P \leq 0.01$; ****$P \leq 0.0001$). Data represent mean ± SEM

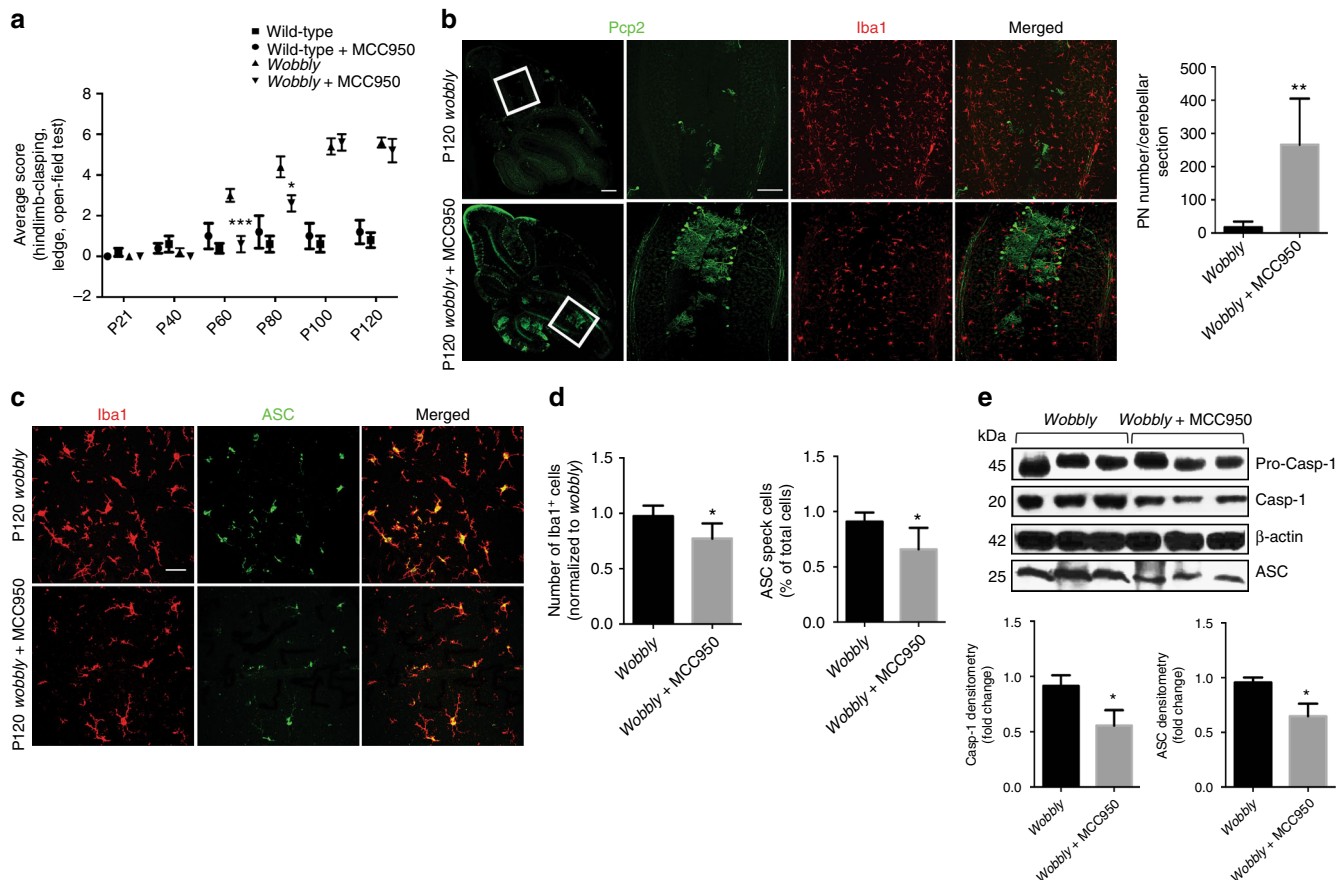

**Fig. 7** MCC950 treatment delays the onset of the *wobbly* phenotype. **a** Analysis of the ataxic phenotype of *wobbly* and MCC950-treated *wobbly* and wild-type mice relative to wild-type controls (*n* = 6 (3 males and 3 females) for each of the genotypes). **b** Pcp2 and Iba1 immuno-staining and PN quantification in P120 *wobbly* mice with and without MCC950 treatment. White rectangles represent magnified areas. **c** Iba1 and ASC immunofluorescence on sagittal cerebellar sections of P120 MCC950-treated and untreated *wobbly* mice. **d** Quantification of microgliosis and ASC specks in (**c**). **e** Western blot analysis and densitometry of cleaved caspase-1 (Casp-1) and ASC in P120 MCC950-treated *wobbly* cerebellar brain lysates relative to the untreated *wobbly* controls. For (**b–e**) *n* = 5 for each of the genotypes; representative images and blots are shown. Scale bars: (**b**) left panel 500 μm; (**b**) middle and right panels, 100 μm; (**c**) 10 μm. Statistical evaluation: (**a**) two-way ANOVA and Sidak's multiple comparisons test; (**b, d, e**) two-tailed *t* test. Statistically significant differences are indicated (*$P \leq 0.05$; **$P \leq 0.01$; ***$P \leq 0.001$). Data represent mean ± SEM

mutant mice. Although reduction of tRNA modification levels is relatively modest, our data is in line with another report on the Elongator mutation that causes familial dysautonomia[43]. In this rare disease, similar levels of reduction in tRNA modifications were observed (29–36% of reduction) in patient-derived samples. Together these observations indicate a scenario, where a certain level of reduction in modification levels causes severe cellular malfunctions, but still permits survival of the patients.

The Elongator-dependent mcm⁵U and ncm⁵U modifications have been shown to be of crucial importance for the fidelity and kinetics of protein synthesis and cotranslational folding dynamics[8,9]. In concordance, we found defects in translational fidelity and protein folding in *wobbly* PNs. Hence, we show that the *Elp6L126Q*-mediated impaired function of the complex likely results in protein misfolding and aggregation that further induces ER-stress and subsequent apoptosis.

Protein aggregation is a common cause of neuronal death shared by various neurological disorders[44,45], as aggregated proteins commonly lose their physiological function and gain undesired toxic properties. As in vast majority of neurodegenerative diseases, cellular pathology is only observed in a specific neuronal subtype in *wobbly* mice, namely PNs. Our transgenic complementation study demonstrates that Elongator is a key regulator of PN integrity and although the observed pathological findings were found to be cell-intrinsic in ataxic mice, the cause of this selectivity remains elusive. In general, neurons are known to be highly sensitive to the presence of misfolded proteins given that they are postmitotic and cannot dilute toxic aggregates by cell division. PNs may be particularly sensitive to deleterious effects of these toxic species as they have an extraordinary high metabolic demand[11,46]. Although codon-dependent regulation of translation by the Elongator complex has been previously reported[6], protein aggregates induced in yeast and worms by Elongator depletion show no specific accumulation of these Elongator-codon enriched proteins. Therefore, the slightly decreased tRNA modification levels primarily might have a large impact on the proper translation of individual trigger proteins, which nucleate and propagate the appearance of large aggregates and induce proteotoxic stress in the context of whole proteome, as suggested by previous studies[8]. The study presented here adds to an emerging consensus that perturbations of the Elongator complex contribute to a range of neurological and neurodevelopmental disorders, including familial dysautonomia[15], amyotrophic lateral sclerosis[17,20], rolandic epilepsy[18], and intellectual disability[16,19]. The mechanism by which specific mutations in different Elongator subunits cause different neuropathologies, remains intriguing and needs to be further clarified.

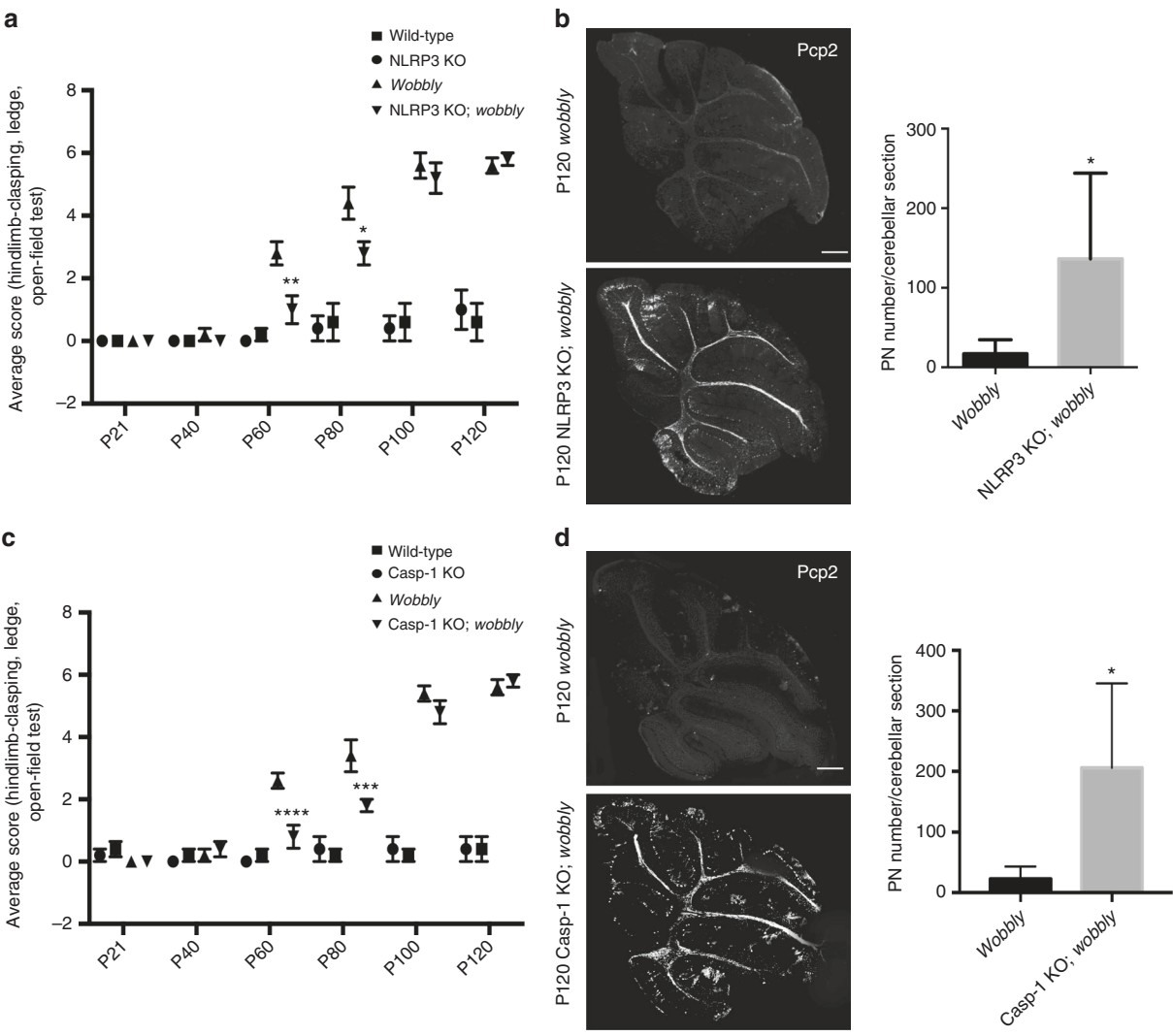

**Fig. 8** Protective effect of NLRP3 and caspase-1 deficiency on the *wobbly* phenotype. **a** Ataxic phenotype analysis of NLRP3 KO, NLRP3 KO; *wobbly* and *wobbly* mice relative to wild-type controls. **b** Pcp2 Immunofluorescence and PN quantification of sagittal cerebellar sections of P120 mutant mice. **c** Behavioral analysis of caspase-1 (Casp-1) KO, Casp-1 KO; *wobbly* and *wobbly* animals relative to wild-type controls. **d** Immunofluorescence of P120 mutant mice sagittal cerebellar sections with Pcp2 antibody. PN quantification is presented. For (**a**) and (**c**) $n = 6$ (3 males and 3 females) for each of the genotypes. For (**b**) and (**d**) $n = 5$ for each of the genotypes; representative images are shown. Scale bars, 500 μm. Statistical evaluation: (**a**, **c**) two-way ANOVA and Sidak's multiple comparisons test; (**b**, **d**) two-tailed $t$ test. Statistically significant differences are indicated ($^{*}P \leq 0.05$; $^{**}P \leq 0.01$; $^{***}P \leq 0.001$; $^{****}P \leq 0.0001$). Data represent mean ± SEM

A number of studies in the past decade have demonstrated that neuroinflammation initiated by the activation of inflammasome complexes in microglia underlies a variety of neuronal phenotypes[23]. Here, we have demonstrated that despite the fundamental mechanism of neuronal death being cell autonomous, activation of the NLRP3 inflammasome contributes significantly to the rate of neuronal loss in *wobbly* mice. Thus, PNs are both dying due to an intrinsic defect and being killed by an extrinsic inflammatory response. Gliosis is a common observation in a number of neurodegenerative conditions and our findings in ataxic mice experimentally confirm an increasing body of evidence proposing a common mechanism of NLRP3-mediated neuroinflammation to underlie and contribute to the pathogenesis of various neurodegenerative diseases[25–28]. Whether NLRP3 activation is a direct consequence of the Elongator complex malfunction in *wobbly* mice or it is triggered by a release of danger-associated molecular patterns (DAMPs) from dying neurons, has to be investigated in the future. It has been previously established that NLRP3 activation is initiated by aggregated proteins, such as Lewy bodies in Parkinson's disease[26], amyloid-β peptides in Alzheimer's disease[25] and prions in prion-related disorders[28]. Thus, the NLRP3-mediated inflammatory cascade is likely to be triggered by the presence of protein aggregates in *wobbly* PNs. Previous studies demonstrated that the NLRP3 activation not only results in cytokine-induced neuronal damage, but also in poor microglial aggregate clearance resulting in accumulation of toxic aggregates that leads to further neuronal demise[47,48]. In order to assess whether aggregates are responsible for the NLRP3-mediated neuroinflammation in cerebellar ataxia or the inflammatory response is induced by neuron-derived DAMPs, different ataxia mouse models with and without the involvement of preoteinopathy need to be compared.

For cerebellar ataxia patients, prognosis is currently bleak and therapies in use are limited to symptomatic treatment with no pharmaceutical intervention available to ameliorate the disease pathology. Data presented here not only define a fundamental

understanding of cerebellar neurodegeneration caused by the Elongator mutation, but also identify a potential therapeutic strategy to substantially delay the course of the disease. Hereditary ataxias display a variety of inheritance modes and the plethora of genetic pathways known to cause these conditions makes it difficult to conceive of a single pan-ataxia therapeutic approach. Though, several studies suggest that the majority of ataxias have some broad commonalities, such as protein aggregation, metabolic deficits, and perturbation in ion channel function[49]. Despite the potentially different biochemical bases that trigger neurodegeneration, a common feature of ataxias is the accompanying induction of neuroinflammation, and in particular microglial activation[50–53]. It is clear that the role of sterile inflammation in neurological diseases is a field that is generating great interest and our study provides the first instance that blocking inflammasome activation can significantly ameliorate neurodegeneration in ataxia by targeting the associated activated gliosis.

## Methods

**Animals and genotyping strategies.** All animal experiments were approved by the University of Queensland Molecular Biosciences Ethics Committee (project license numbers IMB/098/14 and IMB/097/14). Mice were housed under a 12 h light cycle in a specific-pathogen-free climate-controlled facility with food and water provided ad libitum. Genomic DNA for genotyping purposes was obtained from tail tips using QuickExtract DNA extraction solution (Epicenter) as per the manufacturer's instructions. The genetic background of all animals, including *wobbly*, Elp6 KO, transgenic (Pcp2-Elp6-HA), NLRP3 KO and Casp-1 KO mice, was C57BL/6. No gender-related phenotypic differences were observed in all mice strains. Both sexes were found to be fertile with normal life span and body weight. The *wobbly* mutation arose from an ENU-mutagenesis phenotype-driven screen at the Australian Phenomic Facility, the Australian National University. This strain has been archived with the Australian Phenome Bank, ID 4118. Whole-exome sequencing revealed the homozygous *Elp6L126Q* mutation. A custom TaqMan SNP genotyping assay was used to genotype the animals through qPCR reaction based on allelic discrimination. Primers used were: forward 5′-AAACCTGCAG TCACTGTATACGTT-3′ and reverse 5′-CAACAGACTGGGTACTTCCAT-3′, with the cycling conditions: 95 °C for 10 min, followed by 40 cycles of denaturation step at 92 °C for 15 s, and annealing and extension at 60 °C for 90 s. The amplified product of 88 bp was further annealed with either VIC-labeled wild-type *Elp6* sequence 5′-AGGACACCCTGAAGCC-3′ or fluorescein-labeled *wobbly* (*Elp6L126Q*) sequence 5′-AGGACACCCAGAAGCC-3′. For MCC950 treatment studies, *wobbly* mice were dosed orally via drinking water (0.3 mg/ml), from P21 until sacrifice. Elp6 functional KO mice were generated using CRISPR-Cas9 technology with the gRNA sequence: 5′-CCCCAGACAAGACCGAGCAG-3′ (Supplementary Fig. 3a). Primers used for genotyping were: forward 5′-CCCTGAGCCATCTCTTTGGC-3′ and reverse 5′-AGTGTTCCGTGCAACCTG TAA-3′, with an amplified product of 920 bp, further analyzed by Sanger sequencing. Transgenic animals were generated by pro-nuclear injection of the *Pcp2-Elp6-HA* construct into fertilized egg. Pcp2 minigene was previously reported and was a generous gift from Professor Jaroslaw Barski[54]. *Elp6* cDNA was HA-tagged at the C-terminus and inserted into exon 4 of the Pcp2 minigene, followed by linearization of the construct by Hind III and EcoRI digestion, purification and pro-nuclear injection. Transgenic mice were genotyped by PCR reaction using forward 5′-GCCTTGGTATCCTCCTG-3′ and reverse 5′-CCAGGAACACAA GCTGCCCTCTGTCCCG-3′ primers, which generated a 671 bp product in Pcp2-Elp6-HA animals. PCR products were amplified using the following conditions: initial denaturation for 4 min at 94 °C followed by 35 cycles of 94 °C for 30 s, 58 °C for 30 s and 72 °C for 60 s, and a final single extension step of 72 °C for 10 min.

**Behavioral testing.** The simple composite phenotype scoring system used was a modified version of the scoring system for evaluating mouse models for spinocerebellar ataxias[55]. The system was based on the hindlimb clasping, open field and ledge tests. All tests were scored on a scale of 0–2, with a combined total of 0–6 for all three tests. Each test was repeated three times and the mean value of the scores was recorded. Time points chosen for testing mice were P21, P40, P60, P80, P100, and P120.

The ledge test: a mouse was lifted from the cage and placed on the cage's ledge. A score of 0: a mouse walked along the ledge without losing its balance, and lowered itself back into the cage using its paws. A score of 1: a mouse showed signs of tremor and lost its footing while walking along the ledge. A score of 2: severe tremor was observed and the mouse fell off the ledge, or shaked and refused to move despite encouragement.

Hindlimb clasping: a mouse was grasped by the tail near its base and lifted to observe the hindlimb position for 10 s. A score of 0: the hindlimbs were consistently splayed outward, away from the abdomen. A score of 1: both hindlimbs were partially retracted toward the abdomen for more than 50% of the observation time. A score of 2: hindlimbs were entirely retracted and touching the abdomen for more than 50% of the observation time.

Locomotor activity: the test was performed in an arena with walls to prevent escape. A mouse was placed in the center of the open field, and its movement around the arena (horizontal activity) together with the rearing (vertical activity) was recorded for 5 min. A score of 0: a mouse moved normally, with its body weight supported on all limbs, and actively explored its surrounding, showing normal horizontal and vertical activity. A score of 1: tremor was present in both, horizontal and vertical activity of the mouse, and both activities were decreased. A score of 2: a mouse had difficulty to move forward and/or fell down on a side while walking, therefore showing a significant decrease in horizontal activity, while no vertical activity was observed.

P100 animals were additionally tested using rotarod, balance beam and Catwalk system. Mice were trained for three days (three trials a day) prior to recording the final score.

Rotarod: maximum time allowed for the rotarod test was 120 s. The acceleration was from 4 to 40 rpm within 60 s. The latency to fall (time spent on the rotating road) and the speed at the time of drop (rmp) were recorded.

Balance beam: mice were trained to walk from a start platform along a wooden beam elevated 30 cm above the ground, 80 cm long and 3 cm wide gradually narrowing to 1 cm, ending at the goal box. The mouse was placed on the wide end of the beam and allowed to walk the beam distance and enter the goal box. The distance crossed (cm) and the number of missteps was recorded.

Catwalk: mice were allowed to walk across the glass walkway in an unforced manner at least six times a day. Mouse tracks that were straight without any interruption or hesitation were treated as successful runs. Runs with any wall climbing, grooming, and staying on the walkway were not analyzed. Mice that failed the Catwalk training were excluded from the study. An average number of eight replicate crossings made by each mouse was recorded. The Catwalk software was used to analyze crossings that had at least five cycles of complete steps. Gait parameters, such as regularity index and base of support, were collected and compared between groups.

All behavioral analyses were performed by researchers blinded to the genotype of the animals.

**MCC950 treatment studies.** Mice were dosed orally via drinking water (0.3 mg/ml) ad libitum starting at P21 until sacrificed at P120. The MCC950 dose to be used was established in a pilot study where we assessed the penetrance of the drug into the brain tissue of mice at levels above the $IC_{50}$ of the drug ($n = 3$ for MCC950-treated and control animals). The concentration of the drug was measured in blood plasma and brain tissue upon transcardial perfusion with PBS.

**Tissue and embryo collection.** Experimental animals were anaesthetized using Dormitor (1 mg/kg, i.p.) and Zoletil (50 mg/kg, i.p.) and transcardial perfusion was performed with PBS, followed by 4% PFA solution. The brains were dissected and drop-fixed in 4% PFA at 4 °C for 12 h under constant agitation. The following day, brains were washed twice with PBS and left overnight in PBS. Brains were processed in the Leica TP1020 tissue processor over 15 h as per the user's guide and subsequently embedded in paraffin and sectioned at 7 μm either in the transverse or sagittal plain using Leica RM2235 microtome. Sections were transferred to glass slides and dried overnight at 45 °C. Embryos were explanted and placed into cold PBS, followed by 4% PFA fixation for 6 h at 4 °C and subsequent series of PBS washing.

**H&E staining.** Following deparaffinization, slides were stained in Hematoxylin (Sigma Aldrich) for 3 min. The excess of Hematoxylin stain was removed by short immersion of slides in 1% HCl acid solution followed by another short immersion in 0.1% LiCO₃ solution. Samples were then stained with Eosin Y solution (Sigma Aldrich) for 30 s and dehydrated using 70, 90, and 100% ethanol for 30 s each, followed by xylene for 10 min. Slides were mounted with Entellan mounting medium (ProSciTech) and dried for 1 h. Images were obtained using Olympus BX-51 upright bright-field microscope.

**Immunofluorescence.** Upon deparaffinization and hydration, the slides underwent heat-induced antigen retrieval using citrate buffer-based antigen unmasking solution (Abacus) at 100 °C for 10 min. Mouse on Mouse (M.O.M.) blocking reagent (Vector) was used to block endogenous mouse antibody in the tissue section (when a primary antibody was raised in mouse) or bovine serum albumin was used to block unspecific binding of antibodies (when an antibody was not in other specie than mouse). Slides were incubated with primary antibodies: Pcp2 (1:100; sc-49072), Zebrin II (1:50; ab115212), Iba1 (1:400; ab5076), ASC (1:400; Al177), Casp-3 (1:100; ab2302), Hsp70 (1:100; sc-66048), Ub (1:100; ab7780), and HA (1:100; ab9110), followed by AF488, AF594, or AF647-labeled donkey anti-mouse, anti-rabbit or anti-goat IgG antibody (1:250; Invitrogen), and counterstained with DAPI (Sigma Aldrich). Images were captured using Zeiss LSM 710 upright confocal microscope as Z-stacks and presented as the sum of the Z-projection. The number of Pcp2-labeled or Iba1-labeled cells was determined separately in every visible lobule of the vermis and in the hemispheres. For each mouse, three nonadjacent sections (separated by 70 μm) from the region of

vermis were analyzed and the mean value was recorded. Total of five mutant and five control animals was included in each of the studies.

**Electron microscopy**. Excised cerebella were quickly trimmed and immediately immersed in 2.5% glutaraldehyde in cacodylate buffer for 48 h and then postfixed in reduced osmium, en-bloc stained with 2% uranyl acetate and dehydrated through ethanol solutions, before final embedding in Epon812 resin (ProSciTech). Ultra-thin sections were cut on a Leica UC6 Ultra microtome and viewed on a JEOL 1011 electron microscope (JEOL Australasia Pty Ltd) at 80 kV. Images were captured using iTEM software (Soft Imaging System, Olympus).

**In situ hybridization**. Total RNA was extracted from granule neuron precursors isolated from P7 wild-type animals[56] using RNeasy micro kit (QIAGEN) following the protocol supplied. Subsequently, cDNA was obtained using the protocol from SuperScript III first-strand synthesis kit (Thermo Fisher). The probes were amplified using above mentioned PCR conditions. Primers were designed in the 3′UTR region of the *Elp4-6* cDNA: *Elp4* forward 5′-CGACTGCATTTGCCTCCA GACTTGTCAGAC-3′ and reverse 5′-GTTCTACACTCTATGGGGTGT GCCATGCC-3′, *Elp5* forward 5′-GCTTCATGGCCCAGGCTCCATGGG-3′ and reverse 5′-CACACTCATCCTAGCTTGATGCTGCTCCTTGGC-3′, *Elp6* forward 5′-GCCTCACCTGCCTGTTTTTG-3′ and reverse 5′-GTCCCAGTGCCAT GCTTTTG-3′. DIG RNA labeling kit (Roche) was used to synthesize the RNA probes and the probes were purified using the RNA cleanup protocol from the RNeasy mini kit (QIAGEN). In situ hybridization on P21 sagittal brain sections was carried out following the established protocol[57].

**Western blotting**. Mouse cerebellum whole-tissue lysate was prepared by homogenization in radioimmunoprecipitation assay (RIPA) buffer. Protein concentrations were determined using BCA protein assay (Pierce). The XCell SureLock mini-cell electrophoresis system (Thermo Fisher Scientific) was used for SDS-PAGE and wet protein transfer. Proteins were transferred onto nitrocellulose membranes at 25 V for 90 min and detected by immunoblotting using relevant primary and horseradish peroxidase-conjugated secondary antibodies. Peroxidase activity was further detected using SuperSignal West Pico chemiluminescent reagent (Thermo Fisher Scientific). The membrane was exposed to an X-ray film for 30 s–2 min prior to development using X-omat film developer. Densitometric analysis of western blot images was performed using ImageJ software.

**Electrophysiology**. Whole-cell recordings were obtained from PNs in 300 μm-thick sagittal brain slices from P21–26 *wobbly* and wild-type mice. Mice were anesthetized with isoflurane, decapitated, and 300 μm-thick sagittal brain slices prepared in an ice-cold sucrose solution using a vibratome (Leica). Brain slices were continuously perfused with oxygenated aCSF (32 °C) and whole-cell patch-clamp recordings were performed as previously described[58]. Spiking was evoked using current injections applied in increments of 20 pA from −60 to 340 pA. In the case of spontaneous excitatory postsynaptic current (sEPSC) recordings, Purkinje cells were clamped at −60 mV and recorded for 10 min. For electrical stimulation, a theta-glass stimulator (Harvard apparatus glass capillaries) was filled with aCSF and placed in the molecular layer of cerebellar sagittal slices to stimulate parallel fibers. Induced inhibitory postsynaptic current (IPSC) and EPSC were recorded, while holding PNs at −40 and −60 mV, respectively. Input resistance, action potential threshold, amplitude, delay, half width, rise time, sEPSCs, IPSCs, and EPSCs were analyzed offline. No corrections were made for liquid junction potentials.

**tRNA modification analyses**. Total RNA was isolated from ~100 mg cerebellar tissue using TRIzol reagent (Life Technologies). After TRIzol addition, the tissue was homogenized with ceramic beads (Sapphire Bioscience) in tissue homogenizer (Bertin Technologies). Total tRNA was isolated from low melting point agarose gel (Progen) following the protocol for β-agarase I provided by manufacturer (NEB). After completion of gel digestion reaction, tRNA was extracted with water-saturated phenol followed by repetitive chloroform extraction and precipitation by ethanol; 8–10 ng of tRNA was recovered for each sample. tRNA purity and quality was confirmed by 8% urea–PAGE electrophoresis in TBE–buffer. Hydrolysis of tRNA to nucleosides was performed as previously described[59]. A Prominence ultra high performance liquid chromatography (HPLC) system (Shimadzu, Australia) was used to perform reversed phase separation of the samples using a Luna Omega 1.6 μm, Polar-C18 100A column (150 mm × 2.1 mm, Phenomenex, Australia) at a temperature of 35 °C. A 10-μl aliquot of the samples in 0.1% formic acid (aq) was injected onto the HPLC column. The mobile phase consisted of solvent A (0.1% formic acid (aq)) and solvent B (acetonitrile/0.1% formic acid). Chromatographic separation was achieved using a gradient elution program of 0% buffer B for 4 min, 0–20% for 21 min, 20–40% for 2 min, 40–80% buffer B for 1 min, followed by washing with 80% buffer for 2 min and returning to 0% for equilibration of the column before the next sample injection. The flow rate was 200 μl/min. The RP-HPLC column was directly connected to the TurboIonSpray source of the mass spectrometer. Selected reaction ion monitoring (SRM) mass spectrometry experiments were performed on a hybrid quadrupole/linear ion trap 4000 QTRAP MS/MS system (SCIEX, California, USA). All analyses were performed using SRM

positive ion acquisition mode. Mass spectrometer parameters were optimized for targeted ribonucleosides using multiple injections of 0.1–1 ng of uridine, cytidine, adenine, guanine, as well as commercially obtained 5-methylcytidine (Abcam), 7-methylguanosine, inosine, 5-methyluridine and 2′-O-methyladenosine (MetaGene). The obtained retention times were superimposed with the previously published results[59]. Mass Spectrometer Source conditions were: Declustering Potential, DP: 60, Curtain Gas: 40, Collision Gas (CAD): Medium, Ion Spray Voltage: 5300, Temperature GS2: 500, Ion Source Gas 1: 70, Ion Source Gas 2: 80. SRM Transitions: Resolution set to Unit, Unit for both Q1 and Q3 respectively; Dwell Time 130 ms except for U, which was 5 ms, was used for data analysis and quantification. An analytical method was developed for the simultaneous analysis of 33 modified ribonucleosides of which 5 compounds of interest were targeted for quantification (Supplementary Table 2). 5-Methylcytidine served as an internal normalization standard. To confirm the correct retention time, the standard was added to control sample aliquots at 0.01 or 0.1 ng. Analyst 1.6.2 software was used for peak assignment, area calculation and normalization. Corresponding structures and molecular masses were obtained from Modomics database[60].

**Localization of mElp456L126Q within the structure of the Elongator complex**. Elp6 from *S. cerevisiae* (Q04868) was used as a template to identify and align Elp6 sequences from mouse (Q8BK75) and human (AAH00623). Structural comparisons and models were prepared using Phyre2[61] and Modeler[62]. Atomic models of Elp456 (PDB ID 4A8J) and holoElongator were used to identify L126Q and to prepare structural figures using Pymol[63].

**Protein expression and purification**. Coexpression constructs encoding mouse Elongator subunits were designed as previously described for yElp456[9]. Constructs encoding truncations and mutations of mElp4, mElp5, and full-length mElp6 were created using standard quick-change protocol. mElp6 and *mElp6L126Q* were cloned into pETM11 using standard cloning procedures. All constructs were expressed in *E. coli* (BL21 pRARE) after transformation using electroporation. In detail, bacteria were grown in TB at 37 °C until an OD600 of ~1.2, followed by induction with 1 mM IPTG and subsequent incubation at 18 °C for 12–15 h. Bacteria were lysed in 50 mM HEPES (pH 7.5), 300 Mm NaCl, 10 mM imidazole, 1 mM β-mercaptoethanol, 5% (v/v) glycerol, DNAse and protease inhibitors using a homogenizer. The soluble fractions were cleared by centrifugation (70,000×g for 45 min at 4 °C), and proteins were further purified using NiNTA affinity chromatography, followed by size-exclusion chromatography on a 16/600 HiLoad Superdex 200 pg column (GE Healthcare) and/or Superdex 200 Increased (10/300) in 20 mM HEPES (pH 7.5), 150 mM NaCl and 5 mM DTT. Respective fractions were analyzed by SDS-PAGE, pooled and concentrated.

**Thermal shift assay**. Thermal shift assays were performed to monitor protein unfolding using thermofluor technology[64]. Thermofluor assays were conducted in the CFX96 Real-Time System C1000 Touch Thermal Cycler (Biorad). Varying concentrations of protein samples (1–0.25 mg/ml) were incubated with SYPRO Orange and 20 mM HEPES (pH 7.5), 150 mM NaCl, 5 mM DTT buffer. Samples were gradually heated from 4 to 98 °C at a heating rate of 0.2 °C/10 s. The fluorescence intensity was measured at probe specific excitation (470 nm) and emission (570 nm) wavelengths.

**Microscale thermophoresis**. Microscale thermophoresis (MST) experiments were performed to determine binding affinities of Cy5-labeled tRNA^Ala and tRNA^Cys from *S. cerevisiae* (at concentration 210 and 55 nM) and purified Elp456 complexes. Proteins were titrated in a 1:1 dilution series in 20 mM HEPES pH 7.5, 50 mM NaCl, 2 mM DTT, 0.05% Tween, whereas labeled tRNA concentrations stayed constant. Samples were loaded into Monolith™ NT.115 MST Premium Coated Capillaries (NanoTemper Technologies) and measured using a Monolith NT.115 at room temperature (light-emitting diode (LED)/excitation power setting 20%, MST power setting 20%). Data was analyzed using MO. Affinity analysis software at the standard MST on time of 5 s.

**Nano differential scanning fluorimetry**. NanoDSF experiments were performed to determine protein stability employing intrinsic tryptophan or tyrosine fluorescence of purified Elp456 complexes. Proteins were diluted to concentration 100 μg/ml in 20 mM HEPES pH 7.5, 50 mM NaCl, 2 mM DTT. Samples were loaded into Prometheus NT.48 Series nanoDSF Grade High Sensitivity Capillaries (NanoTemper Technologies) and measured using a Prometheus NT.48 at temperature range from 20 to 95 °C (LED/excitation power setting 10%, temperature slope 2 °C/min). Data was analyzed using NanoTemper software, $n = 3$.

**Data analysis and statistics**. Statistical analyses were performed using the GraphPad Prism software V6. To determine statistical significance, the unpaired two-tailed *t* test was performed. For the simple composite phenotype scoring system for cerebellar ataxia and the quantification of PNs, Iba1 and ASC specks at different ages, two-way-analysis of variance (ANOVA) was used. To test for difference of each dependent variable in different age groups, Sidak's test was used.

**Data availability**. Data generated in this publication are available from corresponding authors on reasonable request.

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

## Acknowledgments

The authors acknowledge the facilities, and the scientific and technical assistance of the Australian Phenomics Facility (APF), the Australian National University. The APF is supported by the Australian Phenomics Network (APN). The APN is supported by the Australian Government through the National Collaborative Research Infrastructure Strategy (NCRIS) program. We are very grateful to Jelena Bezbradica Mirkovic and Kate Schroder for providing NLRP3 KO and Caspase-1 KO animals and for their valuable discussion. We also thank Avril Robertson and Matthew Cooper for the gift of MCC950 and Trent Woodruff for advice regarding the administration of MCC950. We acknowledge Ting-Yu Lin and Andrzej Chramiec-Głąbik for providing labeled tRNAs. This work was supported by the POLONEZ1 Grant UMO-2015/19/P/NZ1/02514 from the National Science Centre, Poland and received funding from the European Union's Horizon 2020 research and innovation program under the Marie Skłodowska-Curie grant agreement No. 665778 (M.G. and A.S.-K.) and the First Team grant First TEAM/2016-1/2 from the Foundation for Polish Science (S.G.).

## Author contributions

M.K. designed and performed characterization of the *wobbly* phenotype, Elp6 KO, transgenic complementation, histology and neuroinflammation experiments. M.G. and A.S.-K. cloned constructs, expressed/purified recombinant proteins and performed protein stability assays. M.G. performed and optimized MST assays. M.G. and S.G. designed in vitro experiments and analyzed data. B.K. and M.K. designed and performed zebrin II, PN quantification across cerebellar regions, in situ hybridization, behavioral experiments and cloned mElp456 constructs. S.H., A.T., and P.S. designed and performed electrophysiological recordings. S.M., M.K., A.J., and K.A. performed HPLC/MS tRNA modification analyses. B.W. performed whole-exome sequencing and identified the *Elp6L126Q*. L.A.G. and C.A. provided guidance and critical discussion of data. D.L.B. and J.L.S. performed electron microscopy. S.G. performed structural analyses, prepared figures and contributed to the manuscript text. B.J.W. conceived and managed the project and wrote the manuscript with M.K. and S.G. All authors discussed the results and commented on the manuscript.

## Additional information

**Competing interests:** The authors declare no competing interests.

