## [Peer Review File · Nature Communications]

Reviewers' comments:

Reviewer #1 (Remarks to the Author):

This study by Kojic at all has strengths that include it being the first demonstration of a disorder associated with specific loss of Elp6 function in the mammalian nervous system. However, there are several over-interpretations of data and missing information that temper enthusiasm of this paper being published in Nature Communication.

While it may be the first demonstration of a role of microgliosis and NRLP3 inflammasomes in exacerbating the pathology of a cerebellar ataxia, it is most certainly not the first evidence of neurological dysfunction as a result of mutations in the Elongator complex – since we know from yeast that deletion of any elongator subunit impairs the function of the entire elongator complex. It is definitely not the first study to report a “mechanistic insight into the pathophysiology of a disorder caused by an Elongator mutation” (pg. 2). It is surprising that no mention is made of published studies investigating the pathological consequences in neurons of mutations in Elongator subunits when there has been so much work in this area the past few years: George et al., 2013; Jackson et al., 2014; Laguesse et al., 2015 and 2017; Naftelberg et al., 2016; Ohlen et al., 2017.

Strengths and novelty include showing a role for NRLP3 inflammasome in exacerbating Purkinje neuronal death due to the loss of Elp6 function, rescue of phenotype with PN-specific expression of Elp6 is strong data and shows their phenotype is due to loss of Elp6 in PCs. Another strength is the structural/biochemical analysis of fate of mutated Elp6 is very interesting – i.e. showing loss of protein stability. But those data are interesting more from a structure/function standpoint – since this is not an existing, naturally occurring mutation in patients, it doesn't help us understand a disease pathology since we already understood the mutation was causing a hypomorph or reduction in Elp6 expression. Other strengths include the EM showing occlusions and autophagosomes. Finally, evidence for an inflammatory response is good although it doesn't prove that the inflammation is due to activation of NRLP3 (which is never measured and see concerns about NRLP3 KO below).

Weaknesses:

Mouse model characterization: Ataxia can be caused by loss of proprioception, lesions to the vestibular-spinal tract in addition to damage to the cerebellum. Given the ubiquitous expression of Elp6 throughout the nervous system, and the fact that mutations in Elp3 and Elp1 cause PNS deficits, it would be useful to have done a more systematic analysis of the mouse PNS and CNS to be fully convinced this is truly a model (solely of) cerebellar ataxia.

These Elp6 mouse models are powerful in that they lead to a complete loss of Purkinje cells. That is a dramatic phenotype given the time frame, but I wonder how physiologically relevant it is. That is, certainly PN death is exacerbated by the activated inflammatory response which is typically not as pronounced in the more common slow neurodegenerative disorders like PD or AD. Thus compared to other neurodegenerative diseases, the inflammatory response appears to play a more dominant role in the neurodegeneration here than in the other more common neurodegenerative diseases.

Reduction in tRNA modification may be significant, but still seems quite small. How do we know it is responsible for the protein aggregation? What proteins are aggregated?

Fig. 5b. Quantitative western blots should be conducted for markers of cell stress rather than just snap shots of 1 field.

Fig. 5C is problematic. “The codon content analysis showed that genes expressed in PNs were enriched for the GAA and AAA codons (Fig. 5c), further supporting Elongator to be a key regulator

of translation in these neurons." What are we supposed to compare these numbers to? What defines codon "bias" if we don't know what the codon usage is in e.g. non-neuronal cells, or at least granule cells that don't die due to a loss of Elp6 function?

Fig. 6 – to support the claim that the NLRP3 inflammasome is activated, data would be bolstered by measuring Interleukin 1 β in the cerebellum and whether its levels are reduced in the NLRP3 KO mouse or MCC950 tested mouse.

Fig. 7 – not that impressed with improvement with MCC950 inhibition – seems to rescue about 30% of the Purkinje cells but not impressive reduction in Iba1+ cells nor is behavioral output (hindlimb clasping) reduced, looks like there is a delay in onset only. That means the interpretation of these findings should be tempered. While its suggestive and indicative of a role for the inflammation in the demise of Purkinje cells, its clearly just a component of the cause of PN death.

Fig. 8. Same finding here – an improvement with the NLRP3 KO but not a panacea as described. A rescue of 10% of PNs, a delay in onset of degeneration. The same mild response for deletion of Caspase-1 – slower rate of degeneration, but ultimately, all treatments produce the same degree of degeneration.

Discussion:

This is not a true statement: "Our data represent the first instance where perturbation of Elongator leads to proteinopathy and consequent neurodegeneration." Check the paper by Lageusse et al., 2015.

This is not a valid statement based on the data shown in the paper: "As in vast majority of neurodegenerative diseases, pathology is only observed in specific neurons in wobbly mice, PNs." No presentation of any cell type outside of the cerebellum was included in the paper.

The authors never explain whether activation of the inflammasome is a specific consequence of loss of Elongator function, or just a normal consequence of inducing neuronal death?

Minor:

This is an overstatement – I don't think there is enough mechanistic data to warrant this comment "Over the past decade, there has been an emerging consensus that the Elongator complex is involved in the majority of cellular activities underlying the development and maintenance of the CNS"

This is another exaggerated statement that should be tempered given the lack of definitive direct evidence for what induces the cell death: "Subsequently, we showed that these cells express the endoplasmic reticulum (ER) stress-induced transcription factor CHOP, indicating that apoptosis was induced by an accumulation of misfolded proteins as previously described in various neurodegenerative diseases".

In summary, while there are some new, important findings here:

- Reduction of Elp6 can cause death of purkinje cells
- Reduction of Elp6 can cause protein aggregation
- Reduction of Elp6 can increase levels of Caspase-1 and activate Iba1.
- Reduction in Caspase-1, NLRP3 and MCC950 can reduce the rate of degeneration.

I don't think these data teach us more about how neurons die in response to mutations in Elongator. Nor is there any explanation for why or how the neuronal response to reductions in Elp6

activates the inflammasome (if in fact it is activated).

Reviewer #2 (Remarks to the Author):

The present work describes the analysis of a novel mouse model for late onset cerebellar ataxia associated with microgliosis and progressive degeneration of cerebellar Purkinje neurons. This mouse model suffers from a point mutation in the gene coding Elp6 subunit (Elp6L126Q) of the Elongator complex and leading to expression of an hypomorph variant of Elp6. This likely impairs the stability of Elongator, thereby leading to tRNA side chain modification defects and protein aggregation. This can activate the NLRP3-dependant inflammasome and in this model, the authors showed that prevention of microglial priming by interfering with the NLRP3 inflammasome pathway diminish Purkinje neuron degeneration.

The work is overall well conducted and it highlights novel molecular features responsible for progressive PN degeneration linked to poor translation and proteic stress. However some experiments are need to clarify how MCC950 works to prevent PN degeneration.

Majors

1/The authors claims that the Wobbly mouse only suffers from PN degeneration but do not provide experimental evidences for other brain regions.

Elongator activity is important for proper development of the brain, it is thus critical to show that the « wobbly » does not interferes with neurodevelopmental processess that may secondarily impact on mouse behavior/cognition.

2/Despite reduced AP treshold (slightly significant), there is no clear changes of excitability between WT and Wobbly PN as recordings of EPSCs and IPSCs revealed not significant differences between genotypes (figure S2 and table 1).

3/It is unclear why analysis of tRNA modifications in cerebellar lystates did not revealed any thiolation defects as this modification usually occurs downstream cm5 which is driven by Elongator. Would FACS purifying PN before analysis improve the reading ?

4/The translatome of PN was checked to decipher whether the selective sensitivity to Elp6L126Q would arise from PN specific codon usage. The frequency of GAA and AAA codons is high in PN but how different is it from the one from another cell type that express Elp6 and does not suffer from progressive degeneration?

5/Microglia progressively invade the cerebellum in wobbly mice. However, it is unclear how MCC950 delays neurodegeneration? There is no direct demonstrated that NLRP3 is activated in Wobbly mice and if yes in which cell types exactly. Is the MCC950 neuroprotection occuring by preventing microglia activation and release of proinflammatory cytokines (should be showed) and/or by acting directly on PN? Is it possible to inactiate specifically NLRP3 in microglia ?

Minors

1/The expression of Elp4/5/6 in PN should be pointed by arrows on the figure S4

Reviewer #3 (Remarks to the Author):

In the manuscript entitled "Blocking the NLRP3 inflammasome delays Purkinje neuron degeneration and onset of ataxia", Kojic et al have generated an Elp6 mutant mouse that develops Purkinje cell loss, motor deficits and early electrophysiological deficits in Purkinje cells. The authors show that the mutation affects the stability of the Elongator complex which lead to protein misfolding, ER stress and aggregate accumulation in the Purkinje cells. The authors go on to argue that these effects may be caused by reduced tRNA base modification and Purkinje neuron specific codon preferences (GAA, AAA and CAA). Additionally, the mice show an inflammatory response in the cerebellum and the authors demonstrate that drug or genetic inhibition of the NLRP3 inflammation pathway leads to a delayed onset of the ataxia phenotype.

The authors present a well-written manuscript with a logical flow of experiments that include mechanistic insights into Elp6 function. The authors present a wealth data to support their conclusions, and should be commended for examining the effects of the mutation from protein biochemistry to histopathology and animal behavior. However, there are a number of major concerns regarding over-interpretation of the data and the mechanistic insights that can be drawn. Additionally, it is not at all clear that findings from the wobbly mice will provide insight into the role of inflammation and therapeutic approaches for human cerebellar ataxias. Specific concerns are listed below.

1. The behavioral deficits are presented as a composite assessment score of tests which are all easily subject to observational bias. The authors need to state if the assessments were done by researchers blinded to the genotype of the animals and provide the individual scores for the ledge tests, hindlimb clasping and locomotor activity should be provided. Are any of these significant?

2. The authors need to show data regarding ELP6L126Q heterozygous vs wildtype mice

3. Some additional quantitative motor deficit assessments (e.g. DigiGait, Rotarod, electronic open field, ...) should be added and done in a blinded fashion.

4. The authors emphasize the effects of the Elp6 mutation on the cerebellum but do not provide convincing evidence that other parts of the brain are not affected which would be suspected given that the Elp6 null mice are embryonic lethal. The authors should soften their language somewhat as it is difficult to be confident that other parts of the brain are not also affected.

5. Do the rescue experiments with the PCP2-elp6-HA rescue motor deficits in the wobbly mice?

6. The rationale and description of the experiments in Supplementary Fig. 5e and Fig 4d on page 9 need a better description and explanation. First of all, the graph shown in Suppl Fig. 5e is very difficult to read and no statistical data substantiating that any changes which are not even visible are statistically significant. For Fig 4d, there may be many cellular processes that are affected so the proposed link to what appear to be very modest tRNA modifications in whole cerebellar extracts are correlative at best and do not provide evidence for "a mechanistic link between the death of PN and altered protein synthesis rates and elevated protein aggregates".

7. Similarly, the hypothesis that the PN specificity for neurodegeneration in the wobbly mouse results from codon usage changes is speculative. It is not clear that the modifications that are found in whole cerebellar extracts are actually found in PNs. Some additional data are needed to make this connection and to support the claim that these effects are PN specific. Do gene expression data from PN in wobbly mice support altered expression of genes with enriched codon usage?

8. The authors have presented findings supportive of the general presence of misfolded proteins and computational data suggesting enriched codon usage in PNs genes. What is the argument that the two observations are connected? Do the authors have data linking the misfolded proteins to the computed list of genes? Are the signs of protein misfolding/aggregation restricted to the

Purkinje cells of the cerebellum or found elsewhere in the brain. For the authors to argue for a Purkinje cell specific codon usage enrichment, they need to show enrichment data for another cell type to confirm the enrichment is Purkinje cell specific and not neuron specific.

9. Central to the author's claims is the finding of altered tRNA modifications. However, the overall effect on uridine modification in Figure 4d, while statistically significant, is relatively modest. Can the authors provide additional data or references to show this level of reduced modification is in line with other Elongator mutations?

10. To confirm targeting of microglia as the author assert in the abstract, they should present measures of Iba1 microglia and activated microglia in NLRP3 and Casp-1 KO/wobbly mice in Figure 8.

11. The background of the mice used for the NLRP3 and Casp1 crosses should be described. Were these animals on the same background strain as the wobbly mice? If not this could generate huge differences in background between the triply transgenic animals. A detailed description of these crosses needs to be added to the methods section.

12. The effects of drug and genetic alterations to inflammatory response would not be restricted to cerebellum. Do the authors have any evidence for inflammatory responses and improvements with MCC950 or Nlrp3KO:wobbly mice?

13. The title of the article, abstract and other places within the article overstate the relevance of their findings. Blocking the activation of the NLRP3 inflammasome has not been demonstrated to be a general strategy for ataxias and needs to be more specific and narrowly related to the Elp6/wobbly mice, unless the authors test their findings in multiple ataxia models. The title and statements like "defines a novel therapeutic approach for a severe class of disorders where currently none exist" and "NLRP3 pathway is a putative target for the treatment of ataxias of diverse aetiologies" should be deleted. This would be okay to discuss as a future experiment in the discussion but there is NO data presented to support this general conclusion in this manuscript.

14. Figure 7 is highly speculative and should not be presented as the mechanism underlying neurodegeneration in the wobbly mouse.

15. Image shown in Fig. 7b does not appear to be representative compared to the data shown in Fig. 7a.

16. The number of animals used in most of the behavioral, histological and electrophysiological tests is missing and should be indicated.

17. Additional minor concerns:

- Why is there an asterisk for a p value of 0.49 in Supplemental Table 1 – is this a typo?
- Panels in most of the figures are way too small. The panels and the fonts use for labels needs to be much larger.
- Describe what the red bars in Figure 5 indicate. There is no explanation in the text or the legend.

Reviewer #4 (Remarks to the Author):

In this article Kojic et al evaluate the effects of NLRP3 blockade in a model of neuronal degeneration. The characterisation of the wobbly mouse is sound and nicely pinpoints specific gene

targets. However, the rationale for targeting microglia (or NLRP3) and the actual analysis is rather weak and lacks a thorough study in order to support the authors claims. My main comments can be summarised as below:

Main points

1. As the authors nicely introduce, cerebellar ataxias are a complex group of diseases and, although microgliosis seems associated with the progression of the disease, it seems that the evidence for this being a driving force of it is weak. It could just be a consequence of the over and progressive neurodegeneration. In this sense, what is the evidence for microglia being causative in these diseases? Or, alternatively, how is the current model representative of the human diseases so we can extrapolate findings about microglia? As it stands, it is difficult to gauge the impact of the current findings based on a novel model not yet explored much.
2. The characterisation of the model is excellent, but being this a novel mouse model, the shown characterisation seems too restricted to the cerebellum, and a more comprehensive characterisation is needed (or needs to be show more than just commented in the text). For example, is survival affected? Is there a different phenotype in males vs females? Do any other areas undergo some degree of neurodegeneration or gliosis (in particular hippocampus because of similarities of cell populations)?
3. The characterisation of the gliotic response in the cerebellum of wobbly mice is weak. Only a few micrographs are show, lacking proper analysis. In order to understand the correlation of the neurodegeneration with gliosis, it is necessary to perform a time-course quantification of the number of microglia and astrocytes, as well as the expression of the selected inflammatory markers. Moreover, magnified confocal images are needed in order to better resolve the presence of ASC in glial cells.
4. The authors use a drug MCC950 to block NLRP3, but little information is provided about the action of this compound. What is the precise mode of action? Have you got any data demonstrating this drug is brain penetrant? At what dose?
5. The approaches (pharmacological and genetic) utilised to block NLRP3 seem to cause a transient and modest amelioration. Why is this a transient on behaviour? Moreover, the analysis generally lacks controls, and WT and WT+drug (or genetic controls) should be included, in order to appreciate the magnitude of the effects on PN survival, and also be able to perform adequate statistical comparisons. Currently, comparing wobbly vs treated is incomplete and would provide a biased view of the real effect size.

Minor points:

1. Staining microglia for Iba1 has little meaning, beyond understanding changes in number. It would be highly recommended to stain for activation markers such as MHCII or IL1b, in order to better understand this response to injury.
2. It would be very useful to provide additional behavioural analysis. Being a model of ataxia, it would be expected to observe early changes in rotarod, climbing test or grip tests, and this would reinforce the validity of the model considerably.

Response to Reviewers

We thank the Reviewers for their constructive comments. As a result we have added significant new data and provided further clarification that we believe has resulted in a significantly improved manuscript. In addition to the detailed response below, Appendix 1 contains an overall summary of the new data added.

Reviewer #1 (Remarks to the Author):

This study by Kojic at all has strengths that include it being the first demonstration of a disorder associated with specific loss of Elp6 function in the mammalian nervous system. However, there are several over-interpretations of data and missing information that temper enthusiasm of this paper being published in Nature Communication.

While it may be the first demonstration of a role of microgliosis and NRLP3 inflammasomes in exacerbating the pathology of a cerebellar ataxia, it is most certainly not the first evidence of neurological dysfunction as a result of mutations in the Elongator complex – since we know from yeast that deletion of any elongator subunit impairs the function of the entire elongator complex. It is definitely not the first study to report a “mechanistic insight into the pathophysiology of a disorder caused by an Elongator mutation” (pg. 2). It is surprising that no mention is made of published studies investigating the pathological consequences in neurons of mutations in Elongator subunits when there has been so much work in this area the past few years: George et al., 2013; Jackson et al., 2014; Laguesse et al., 2015 and 2017; Naftelberg et al., 2016; Ohlen et al., 2017.

1. We have rephrased this sentence in the revised version of the manuscript on page 2 (abstract) to “Our data provide a mechanistic insight into the pathophysiology of a cerebellar ataxia caused by an Elongator mutation, substantiating the increasing body of evidence that alterations of this complex are broadly implicated in the onset of a number of diverse neurological disorders.”. Recent studies investigating the pathological consequences in neurons affected by Elongator mutations are now referenced on page 3 in the revised version of the manuscript. “Over the past decade, a number of studies have showed that the Elongator complex is involved in various cellular activities that govern the development and maintenance of the nervous system¹⁰⁻¹⁴”.

Strengths and novelty include showing a role for NRLP3 inflammasome in exacerbating Purkinje neuronal death due to the loss of Elp6 function, rescue of phenotype with PN-specific expression of Elp6 is strong data and shows their phenotype is due to loss of Elp6 in PCs. Another strength is the structural/biochemical analysis of fate of mutated Elp6 is very interesting – i.e. showing loss of protein stability. But those data are interesting more from a structure/function standpoint – since this is not an existing, naturally occurring mutation in patients, it doesn't help us understand a disease pathology since we already understood the mutation was causing a hypomorph or reduction in Elp6 expression. Other strengths include the EM showing occlusions and autophagosomes. Finally, evidence for an inflammatory response is good although it doesn't prove that the inflammation is due to activation of NRLP3 (which is never measured and see concerns about NRLP3 KO below).

Weaknesses:

Mouse model characterization: Ataxia can be caused by loss of proprioception, lesions to the vestibular-spinal tract in addition to damage to the cerebellum. Given the ubiquitous expression of Elp6 throughout the nervous system, and the fact that mutations in Elp3 and Elp1 cause PNS deficits, it would be useful to have done a more systematic analysis of the mouse PNS and CNS to be fully convinced this is truly a model (solely of) cerebellar ataxia.

2. We have performed necropsic analysis of various organs of *wobbly* mice, including testis/epididymis, penis/preputial gland, prostate/seminal vesicles, uterus/ovaries/vagina, urinary bladder, liver/gall bladder, cecum, colon, spleen/pancreas, mesenteric lymph node, stomach, duodenum, jejunum, ileum, kidney/adrenal, salivary glands/lymph nodes, thymus, lungs, heart, skin, eyes, brain, spinal cord, skeletal muscle, skeletal tissue/hind leg. With the exception the initially described PN degeneration and gliosis in the molecular layer of the cerebellum of *wobbly* mice we found no pathological changes including all PNS and CNS structures. The complete list of analysed tissues and organs is now listed in the figure legend to the new Supplementary Figure 3. Given that the majority of ataxias are associated with lesions in the spinal cord, we also analysed the *wobbly* mouse spinal cord and screened for potential cellular defects in addition to observed defects in the cerebellum. No signs of neurodegeneration, gliosis or any lesions were detected in the spinal cord, nor in any other part of the brain, which lead us to the conclusion that the *wobbly* mutant is a new mouse model for cerebellar ataxia with neurodegenerative changes restricted solely to the cerebellum. We have included data on gross morphology of the brain and spinal cord in *wobbly* mice in the revised version of the manuscript in the newly added Supplementary Fig. 3.

These Elp6 mouse models are powerful in that they lead to a complete loss of Purkinje cells. That is a dramatic phenotype given the time frame, but I wonder how physiologically relevant it is. That is, certainly PN death is exacerbated by the activated inflammatory response which is typically not as pronounced in the more common slow neurodegenerative disorders like PD or AD. Thus compared to other neurodegenerative diseases, the inflammatory response appears to play a more dominant role in the neurodegeneration here than in the other more common neurodegenerative diseases.

Reduction in tRNA modification may be significant, but still seems quite small. How do we know it is responsible for the protein aggregation? What proteins are aggregated?

3. The magnitude of change we observe is completely consistent with that observed in published data from familial dysautonomia patients who are defective for Elp1 (Karlsborn et al, 2014, and see below). We would also like to re-emphasise that it is clear from our data that the Elp6L126Q mutation is not a null mutation at either the phenotypic or molecular level – our Elp6KO results in early embryonic lethality. The quantum of tRNA modification reduction we observe is not surprising given that although the *Elp6L126Q* mutation negatively affects the stability of the complex, it does not fully disable its assembly and function (Supplementary Figure 8). Our unpublished data (Glatt lab) on the yeast Elongator complex also indicate that Elp456 is involved in removing modified tRNAs from the complex rather than being involved in the delivery of tRNA or in the modification reaction itself. These data suggest that destabilisation of the Elp456 ring actually affects the overall

reaction efficiency rather than inhibiting the modification reaction. Therefore, our observation would support a model where the *wobbly* mutation leads to non-catastrophic changes in the activity of Elongator and consequently a significant but subtle decrease in tRNA modification, which affects PNs specifically possibly due to their high metabolic demand. We now further elaborate on this theme on page 18: “Although reduction of tRNA modification levels is relatively modest, our data are in line with another report on the Elongator mutation that causes familial dysautonomia⁴⁴. In this disease, similar levels of reduction in tRNA modifications were observed (29-36% reduction) in patient-derived samples. Together these observations indicate a scenario, where a certain level of reduction in modification levels causes severe cellular malfunctions, but still permits survival of the patients.”.

The loss of these modifications in a subset of tRNAs has been shown to lead to ribosome pausing at their cognate codons and protein aggregation in both yeast and *C. elegans* (Nedialkova and Leidel, 2015). Based on these studies and our observations that demonstrate extensive autophagy, ubiquitination and the upregulation of the molecular chaperone Hsp70 in *wobbly* PNs (markers of protein misfolding and aggregation; similarly shown in Lee et al, 2006), we propose a link between the protein aggregation in *wobbly* neurons to tRNA hypomodification. This is consistent with the data of Lee et al 2006 (Nature 443; 50-55) who demonstrated a link between amino-acylated tRNA regulation, protein aggregation and neurodegeneration. We are not able to show which specific proteins are aggregated in *wobbly* PNs. We have previously attempted to purify *wobbly* PNs (*wobbly* Pcp2-GFP strain) in order to identify aggregated proteins in these neurons using FACS. However, this method is only efficient in young mice (P7-P14), where PNs are smaller and with shorter dendritic trees (our data, and also see Tomomura et al, 2001). In older mice, FACS-sorting PNs results in their death, likely due to the mechanical damage of their interconnected dendrites sustained in the process of tissue dissociation. Protein aggregates are observed in *wobbly* PNs from the time point P40 onwards, hence, we have not been able to physically isolate the relevant population of neurons for the purpose of identifying the content of the protein aggregates. In addition, it has been shown that in yeast highly abundant proteins (without specific codon enrichment) constitute the majority of the observed aggregates (Nedialkova and Leidel, 2015). Therefore, we believe that we would not necessarily find any correlation between aggregated proteins and ORFs enriched in Elongator codons. In summary, the detection of specific PN proteins affected by Elongator mutation is not technically feasible and in any event such an analysis is likely to be confounded by other factors.

Fig. 5b. Quantitative western blots should be conducted for markers of cell stress rather than just snap shots of 1 field.

4. We have not been able to detect ER stress and protein aggregation markers by western blot due to a high number of granule neurons (GNs) present in the tissue samples in comparison to PNs (200:1; Williams and Herrup, 1988). In the case of the *wobbly* brain, the ratio is even higher and further shifted towards GNs as the PNs degenerate and FACS-based enrichment is technically challenging (as described above). Thus, the PN protein detection/signal is masked by GNs. Nevertheless, we

have clearly and reproducibly demonstrated upregulation of these markers by immunofluorescence in mutant mice, whilst control wild-type littermates do not demonstrate any such expression. (n = 5 per genotype). This approach is identical to that used by Lee et al, 2006 (Nature 443; 50-55).

Fig. 5C is problematic. “The codon content analysis showed that genes expressed in PNs were enriched for the GAA and AAA codons (Fig. 5c), further supporting Elongator to be a key regulator of translation in these neurons.” What are we supposed to compare these numbers to? What defines codon “bias” if we don't know what the codon usage is in e.g. non-neuronal cells, or at least granule cells that don't die due to a loss of Elp6 function?

5. Following the recommendation of the reviewer, we computed the codon usage in GNs using RNA-seq data from Sharma et al, 2015. We found no significant difference in the codon usage of PNs compared to GNs and we therefore removed PN codon usage data from the manuscript. We thank the reviewer for bringing this to our attention. We now elaborate on page 19: “Although codon-dependent regulation of translation by the Elongator complex has been previously reported⁶, protein aggregates induced in yeast and worms by Elongator depletion show no specific accumulation of these Elongator-codon enriched proteins. Therefore, the slightly decreased tRNA modification levels primarily might have a large impact on the proper translation of individual trigger proteins, which nucleate and propagate the appearance of large aggregates and induce proteotoxic stress in the context of whole proteome, as suggested by previous studies⁸.”.

Fig. 6 – to support the claim that the NLRP3 inflammasome is activated, data would be bolstered by measuring Interleukin 1 in the cerebellum and whether its levels are reduced in the NLRP3 KO mouse or MCC950 tested mouse.

6. We attempted to analyse Interleukin-1 β by western blot, but we had no success likely due to Interleukin-1 β being degraded during brain extraction or tissue lysis process and/or being within relatively low concentration levels. However, as detailed in the manuscript we have clearly demonstrated that levels of ASC nucleation and activation of caspase-1 (both represent markers of the inflammasome activity) are reduced in NLRP3 KO and MCC950-treated *wobbly* mice. It is clear that the NLRP3 inflammasome is activated in *wobbly* mice.

Fig. 7 – not that impressed with improvement with MCC950 inhibition – seems to rescue about 30% of the Purkinje cells but not impressive reduction in Iba1+ cells nor is behavioral output (hindlimb clasping) reduced, looks like there is a delay in onset only. That means the interpretation of these findings should be tempered. While its suggestive and indicative of a role for the inflammation in the demise of Purkinje cells, its clearly just a component of the cause of PN death.

7. We completely agree – it is clear from our data that the PNs are both “dying” from a cell autonomous defect and “being killed” by the concomitant NLRP3-mediated inflammatory response. As explained in the manuscript, neuroinflammation is only a contributing factor to the observed neurodegeneration (page 20). Introduction and discussion chapters are revised in the new version of the manuscript to clarify this further for the general readership and the interpretation of data is tempered and further clarified. Nevertheless, we would like to re-emphasise that our results demonstrate

that inflammation (NLRP3 pathway specifically) can contribute to neurodegeneration in cerebellar ataxia and that targeting neuroinflammation can delay the onset of the disease. In the long term this might represent a valuable clinical intervention point where currently there are none, subject to a greater clarification of the role of NLRP3 activation in human cerebellar ataxia.

Fig. 8. Same finding here – an improvement with the NLRP3 KO but not a panacea as described. A rescue of 10% of PNs, a delay in onset of degeneration. The same mild response for deletion of Caspase-1 – slower rate of degeneration, but ultimately, all treatments produce the same degree of degeneration.

8. Please refer to the response to comment 7.

Discussion:

This is not a true statement: “Our data represent the first instance where perturbation of Elongator leads to proteinopathy and consequent neurodegeneration.” Check the paper by Lageusse et al., 2015.

9. We agree with the reviewer and have rephrased the statement and now focus on ataxias: “Our data represent the first instance where perturbation of Elongator leads to proteinopathy and consequent neurodegeneration in cerebellar ataxia.”

This is not a valid statement based on the data shown in the paper: “As in vast majority of neurodegenerative diseases, pathology is only observed in specific neurons in wobbly mice, PNs.” No presentation of any cell type outside of the cerebellum was included in the paper.

10. We have checked a comprehensive number of different cell types and tissues and included sections of the CNS and spinal cord as a representative of all experiments performed. Please also refer to the response to comment 2, and Supplementary Fig. 3.

The authors never explain whether activation of the inflammasome is a specific consequence of loss of Elongator function, or just a normal consequence of inducing neuronal death?

11. This is now discussed in the revised version of the manuscript on page 20: “Whether NLRP3 activation is a direct consequence of the Elongator complex malfunction in *wobbly* mice or it is triggered by a release of danger-associated molecular patterns (DAMPs) from dying neurons, is yet to be investigated. It has been previously established that NLRP3 activation is initiated by aggregated proteins, such as Lewy bodies in Parkinson’s disease²⁶, amyloid- β peptides in Alzheimer’s disease²⁵ and prions in prion-related disorders²⁸. Thus, the NLRP3-mediated inflammatory cascade is likely to be triggered by the presence of protein aggregates in *wobbly* PNs. Previous studies have demonstrated that the NLRP3 activation not only results in cytokine-induced neuronal damage, but also in poor microglial aggregate clearance resulting in accumulation of toxic aggregates that leads to further neuronal demise^{48,49}.”

Minor:

This is an overstatement – I don't think there is enough mechanistic data to warrant this comment “Over the past decade, there has been an emerging consensus that the Elongator complex is involved in the majority of cellular activities underlying the development and maintenance of the CNS”

12. This statement is changed in the revised version of the manuscript to: “Over the past decade, a number of studies have showed that the Elongator complex is involved in various cellular activities that govern the development and maintenance of the nervous system¹⁰⁻¹⁴.”.

This is another exaggerated statement that should be tempered given the lack of definitive direct evidence for what induces the cell death: “Subsequently, we showed that these cells express the endoplasmic reticulum (ER) stress-induced transcription factor CHOP, indicating that apoptosis was induced by an accumulation of misfolded proteins as previously described in various neurodegenerative diseases”.

13. This statement is changed in the revised version of the manuscript to: “Subsequently, we showed that these cells express the endoplasmic reticulum (ER) stress-induced transcription factor CHOP indicating that apoptosis was likely induced by an accumulation of misfolded proteins (unfolded protein response - UPR) as previously described in various other neurodegenerative diseases³⁸.” The link between the UPR, ER stress and apoptosis in neurodegenerative disease is reviewed in Hetz and Mollereau, 2014 (Nat. Rev. Neurosci. 15; 233-49).

In summary, while there are some new, important findings here:

- Reduction of Elp6 can cause death of purkinje cells
- Reduction of Elp6 can cause protein aggregation
- Reduction of Elp6 can increase levels of Caspase-1 and activate Iba1.
- Reduction in Caspase-1, NLRP3 and MCC950 can reduce the rate of degeneration.

I don't think these data teach us more about how neurons die in response to mutations in Elongator. Nor is there any explanation for why or how the neuronal response to reductions in Elp6 activates the inflammasome (if in fact it is activated).

14. We agree that this manuscript contains important new findings. As discussed (see point 7) consistent with the known function of the Elongator complex and the molecular evidence we observe in *wobbly* and Elp6-corrected *wobbly* mice it is most likely that the mechanism of neuronal death is proteinopathy. The inflammasome in microglia is activated in *wobbly* mice (and not in corrected animals). The inflammasome is typically activated by intracellular ligands and so it is indeed of interest to identify which specific signal is being released from PNs that specifically activates Nlrp3, as distinct from promoting the microgliosis (or it may do both). Unfortunately, ex vivo and in vivo systems are not available to us to address this question - the purification of intact functioning Purkinje Neurons (PN) is an ongoing issue (see comment #3).

Reviewer #2 (Remarks to the Author):

The present work describes the analysis of a novel mouse model for late onset cerebellar ataxia associated with microgliosis and progressive degeneration of cerebellar Purkinje neurons. This mouse model suffers from a point mutation in the gene coding Elp6 subunit (Elp6L126Q) of the Elongator complex and leading to expression of an hypomorph variant of Elp6. This likely impairs the stability of Elongator, thereby leading to tRNA side chain modification defects and protein aggregation. This can activate the NLRP3-dependant inflammasome and in this model, the authors showed that prevention of microglial priming by interfering with the NLRP3 inflammasome pathway diminish Purkinje neuron degeneration. The work is overall well conducted and it highlights novel molecular features responsible for progressive PN degeneration linked to poor translation and proteic stress. However some experiments are need to clarify how MCC950 works to prevent PN degeneration.

Majors

1/The authors claims that the Wobbly mouse only suffers from PN degeneration but do not provide experimental evidences for other brain regions.

Elongator activity is important for proper development of the brain, it is thus critical to show that the « wobbly » does not interferes with neurodevelopmental processess that may secondarily impact on mouse behavior/cognition.

15. We have analysed a comprehensive range of tissues and cell types - please refer to the detailed response to comment 2 and Supplementary Fig. 3.

2/Despite reduced AP treshold (slightly significant), there is no clear changes of excitability between WT and Wobbly PN as recordings of EPSCs and IPSCs revealed not significant differences between genotypes (figure S2 and table 1).

16. Based on data presented, there is no significant difference in the rise and decay time, frequency and amplitude of EPSCs and IPSCs. However, *wobbly* PNs are characterized by a higher number of IPSCs and lower number of EPSCs in comparison to wild-type (Supplementary Fig. 5 in the revised manuscript), showing a change in excitation/inhibition balance. The overall biophysics of EPSCs and IPSCs are not different showing no major synaptic changes, however, mutant PNs do show altered passive and active membrane properties as presented in Fig. 1d-f. Together these changes will have a major impact on PN function.

3/It is unclear why analysis of tRNA modifications in cerebellar lystates did not revealed any thiolation defects as this modification usually occurs downstream cm5 which is driven by Elongator. Would FACS purifying PN before analysis improve the reading ?

17. As described previously, FACS enrichment of PNs does not yield viable PNs and is therefore technically not feasible. Please refer to the response to comment 3 in this regard.

The commonly accepted opinion in the field is that the thiolated species are very hard to detect using MS, due to ionization issues (Alings et al, 2015). As only 3 (4 in

humans) of the 11 tRNAs subject to Elongator activity are in fact thiolated, the changes are expected to be even more subtle and difficult to detect in the background of GNs. Nevertheless, we fully agree with the reviewer that the quantification of mcm5s2U levels would be very interesting in PNs. Therefore, we initially have used also synthetic mcm5s2U nucleotides in our MS calibration runs, but were unable to detect significant signals for this modification in our samples. It should be mentioned that most available studies are currently focusing on the isolation of individual tRNA iso-acceptor species that carry thio-modifications before subsequent quantitative HPLC, MS or Northern blot analyses from model systems (e.g. yeast). As the available biological starting material in this study is very limited, we believe that the necessary isolation of individual tRNA species from PNs is technically challenging and out of the scope of this study.

4/The translome of PN was checked to decipher whether the selective sensitivity to Elp6L126Q would arise from PN specific codon usage. The frequency of GAA and AAA codons is high in PN but how different is it from the one from another cell type that express Elp6 and does not suffer from progressive degeneration?

18. Please refer to the response to comment 5.

5/Microglia progressively invade the cerebellum in wobbly mice. However, it is unclear how MCC950 delays neurodegeneration? There is no direct demonstration that NLRP3 is activated in Wobbly mice and if yes in which cell types exactly. Is the MCC950 neuroprotection occurring by preventing microglia activation and release of proinflammatory cytokines (should be shown) and/or by acting directly on PN? Is it possible to inactivate specifically NLRP3 in microglia ?

19. We showed that there is an inflammasome activation in the *wobbly* mice by showing ASC nucleation and Casp-1 activation (western blot and IF) and we also showed by IF that ASC is nucleated in primed microglia, which is consistent with previous data that within the brain the NLRP3 inflammasome is specifically active in microglia. (Gustin et al, 2015; refer to Fig. 7 in the revised version of the manuscript, and reviewed in Song et al, 2017). We have not observed ASC nucleation in any other type of cerebellar cells. This demonstrates the presence of an inflammasome activation in the *wobbly* brain and through the use of a specific inhibitor of NLRP3 (MCC950) and observing the amelioration of the phenotype (refer to Fig. 7 in the revised manuscript), we demonstrate the link between NLRP3 inflammasome activation in microglia and its contribution to the *wobbly* phenotype. Furthermore, we confirmed these findings by using double mutant Casp1 null/*wobbly* and Nlrp3 null/*wobbly* animals in which inhibition of both inflammasome components gave very similar results, and indistinguishable from the data obtained by treatment with MCC950 alone (refer to Fig. 8 and Supplementary Fig. 10 in the revised manuscript).

The MCC950 neuroprotection likely occurs by preventing microglial priming as MCC950 blocks the NLRP3 inflammasome activation shown to be present in microglia only (as described above; the mode of action of MCC950 is described in Coll et al, 2015 and the reference is provided in the manuscript on page 14). We were not able to detect cytokines from cerebellar lysates as detailed in the response to

comment 6. Nonetheless, we have clearly demonstrated that MCC950 reduces the microglial priming and inflammasome activity (refer to Fig. 7).

We could inactivate the NLRP3 in microglia specifically (using the Iba-1 promoter for instance) however for the 2+ years it would take for us to perform this experiment we do not believe that it would significantly enhance the conclusions of this paper. As has been detailed above, previous studies have already demonstrated that within the brain the NLRP3 inflammasome is specifically active in microglia. By our own observations (this paper) the only evidence we observe of inflammasome activation in the cerebellum is in microglia, not PNs.

Minors

1/The expression of Elp4/5/6 in PN should be pointed by arrows on the figure S4

20. Arrows are added in the revised version of this figure (please refer to Supplementary Fig. 7 in the revised manuscript).

Reviewer #3 (Remarks to the Author):

In the manuscript entitled “Blocking the NLRP3 inflammasome delays Purkinje neuron degeneration and onset of ataxia”, Kojic et al have generated an Elp6 mutant mouse that develops Purkinje cell loss, motor deficits and early electrophysiological deficits in Purkinje cells. The authors show that the mutation affects the stability of the Elongator complex which lead to protein misfolding, ER stress and aggregate accumulation in the Purkinje cells. The authors go on to argue that these effects may be caused by reduced tRNA base modification and Purkinje neuron specific codon preferences (GAA, AAA and CAA). Additionally, the mice show an inflammatory response in the cerebellum and the authors demonstrate that drug or genetic inhibition of the NLRP3 inflammation pathway leads to a delayed onset of the ataxia phenotype.

The authors present a well-written manuscript with a logical flow of experiments that include mechanistic insights into Elp6 function. The authors present a wealth data to support their conclusions, and should be commended for examining the effects of the mutation from protein biochemistry to histopathology and animal behavior. However, there are a number of major concerns regarding over-interpretation of the data and the mechanistic insights that can be drawn. Additionally, it is not at all clear that findings from the wobbly mice will provide insight into the role of inflammation and therapeutic approaches for human cerebellar ataxias. Specific concerns are listed below.

1. The behavioral deficits are presented as a composite assessment score of tests which are all easily subject to observational bias. The authors need to state if the assessments were done by researchers blinded to the genotype of the animals and provide the individual scores for the ledge tests, hindlimb clasping and locomotor activity should be provided. Are any of these significant?

21. All assessment was done by researchers being blinded to the genotype of the animals and this statement is provided in the revised version of the manuscript on page 34.

Individual analyses for the ledge test, hindlimb clasping and locomotor activity are provided in the revised version of the manuscript in Supplementary Fig. 1.

2. The authors need to show data regarding ELP6L126Q heterozygous vs wildtype mice

22. We have provided these data in the revised version of the manuscript in Fig. 1a, Supplementary Fig. 1 and Supplementary Fig. 2.

3. Some additional quantitative motor deficit assessments (e.g. DigiGait, Rotarod, electronic open field, ...) should be added and done in a blinded fashion.

23. We have provided these data in the revised version of the manuscript in Supplementary Fig. 2. The additional behavioral tests include rotarod, balance beam and Catwalk system.

4. The authors emphasize the effects of the Elp6 mutation on the cerebellum but do not provide convincing evidence that other parts of the brain are not affected which would be suspected given that the Elp6 null mice are embryonic lethal. The authors should soften their language somewhat as it is difficult to be confident that other parts of the brain are not also affected.

24. We apologise for not adding the analyses of other organs and cell types in the initial version, and we understand that in the absence of these data some of our statements might have been interpreted as oversimplification. These data have now been added. Please refer to the detailed response to comment 2 and Supplementary Fig. 3 in the revised manuscript.

5. Do the rescue experiments with the PCP2-elp6-HA rescue motor deficits in the wobbly mice?

25. The introduction of PCP2-Elp6-HA fully restores the gait abilities. We provide these data in the revised version of the manuscript in Fig. 3b.

6. The rationale and description of the experiments in Supplementary Fig. 5e and Fig 4d on page 9 need a better description and explanation. First of all, the graph shown in Suppl Fig. 5e is very difficult to read and no statistical data substantiating that any changes which are not even visible are statistically significant. For Fig 4d, there may be many cellular processes that are affected so the proposed link to what appear to be very modest tRNA modifications in whole cerebellar extracts are correlative at best and do not provide evidence for “a mechanistic link between the death of PN and altered protein synthesis rates and elevated protein aggregates”.

26. Supplementary Fig. 5 is revised, please refer to Supplementary Fig. 8 in the revised version of the manuscript. We have simplified the graph presenting MST data in Supplementary Fig. 8e by directly comparing tRNA binding of mouse Elp456 (wild-type) and Elp456L126Q (mutant). We have excluded curves for yeast Elp456, as it seemed to be rather distracting and confusing for the readers. Kd values with corresponding confidence presented as standard deviation values are indicated on the graph. The initial ranges of the standard deviation of calculated Kds show a clear difference between the wild-type and the mutant Elp6 (as shown above the graphs).

To further support our finding, we have included additional analyses of another tRNA molecule (tRNA^{Cys}) which showed a very similar difference in binding capacity of mutated Elp456 by Elongator. Comparison of two tRNA types, a modifiable (tRNA^{Ala}) and non-modifiable (tRNA^{Cys}) by Elongator, showed that the Elongator complex in *wobbly* mice has a decreased capacity to bind tRNA molecules. We have also performed additional nanoDSF experiments (new panel f in Supplementary Fig. 8) that show a clear increase in the amount of aggregated Elp456 protein in the L126Q samples (in agreement with our protein stability data). Due to the recent data, the affinities might artificially be lowered by increased levels of aggregated protein in the mutant samples and we would refer to it as “reduced binding capacities” rather than “reduced binding affinities” (page 10).

The statement regarding tRNA modification providing evidence for “a mechanistic link between the death of PN and altered protein synthesis rates and elevated protein aggregates” is removed from the manuscript. The link between reduced tRNA modification levels as a consequence of *Elp6L126Q* and protein aggregation and PN death, is discussed on pages 18 and 19 in the revised version of the manuscript.

7. Similarly, the hypothesis that the PN specificity for neurodegeneration in the *wobbly* mouse results from codon usage changes is speculative. It is not clear that the modifications that are found in whole cerebellar extracts are actually found in PNs. Some additional data are needed to make this connection and to support the claim that these effects are PN specific. Do gene expression data from PN in *wobbly* mice support altered expression of genes with enriched codon usage?

27. Please refer to the response to comment 5 in regards to the PN codon usage. We do not know if there is a change in gene expression in *wobbly* PNs. This is difficult to determine as we are not able to isolate these neurons and perform RNA-seq or other quantitative transcriptome analyses (please refer to the response to comment 3).

8. The authors have presented findings supportive of the general presence of misfolded proteins and computational data suggesting enriched codon usage in PNs genes. What is the argument that the two observations are connected? Do the authors have data linking the misfolded proteins to the computed list of genes? Are the signs of protein misfolding/aggregation restricted to the Purkinje cells of the cerebellum or found elsewhere in the brain. For the authors to argue for a Purkinje cell specific codon usage enrichment, they need to show enrichment data for another cell type to confirm the enrichment is Purkinje cell specific and not neuron specific.

28. Please refer to the detailed response to comments 4 and 5.

9. Central to the author’s claims is the finding of altered tRNA modifications. However, the overall effect on uridine modification in Figure 4d, while statistically significant, is relatively modest. Can the authors provide additional data or references to show this level of reduced modification is in line with other Elongator mutations?

29. This is discussed on page 18 in the revised manuscript: “Although reduction of tRNA modification levels is relatively modest, our data are in line with another report on the Elongator mutation that causes familial dysautonomia⁴⁴. In this disease, similar levels of reduction in tRNA modifications were observed in patient-derived samples (29-

36% reduction).”. It is also important to note that familial dysautonomia is caused by more drastic mutation in the *Elp1* than we observe in the *wobbly* mutant. Please also refer to the previous response to comment 3.

10. To confirm targeting of microgliosis as the author assert in the abstract, they should present measures of Iba1 microglia and activated microglia in NLRP3 and Casp-1 KO/*wobbly* mice in Figure 8.

30. We have included these data in the revised version of the manuscript in Supplementary Fig. 10 and we specifically refer to this new data on page 15: “Microgliosis and ASC nucleation were found to be reduced to a lesser extent in caspase-1 KO; *wobbly* animals in comparison to NLRP3 KO; *wobbly* and MCC950-treated *wobbly* mutants, which is expected given that the inflammatory cascade in these mutants is inhibited downstream of ASC nucleation³⁰.”.

11. The background of the mice used for the NLRP3 and Casp1 crosses should be described. Were these animals on the same background strain as the *wobbly* mice? If not this could generate huge differences in background between the triply transgenic animals. A detailed description of these crosses needs to be added to the methods section.

31. We now provide detailed information on page 31 in the revised manuscript: “The genetic background of all animals, including *wobbly*, *Elp6* KO, transgenic (*Pcp2-Elp6-HA*), NLRP3 KO and Casp-1 KO mice, was C57BL/6.”.

12. The effects of drug and genetic alterations to inflammatory response would not be restricted to cerebellum. Do the authors have any evidence for inflammatory responses and improvements with MCC950 or *Nlrp3*KO:*wobbly* mice?

32. Other parts of the *wobbly* brain are not affected by neuroinflammation and no microgliosis is observed in other parts of the CNS. “The appearance of these inflammatory markers is confined to the cerebellum and strictly associated with PN degeneration.”, as stated on page 13 in the revised manuscript (in addition to discussion on other parts of the CNS being unaffected in mutant animals on page 6 and data provided in Supplementary Fig. 3). Thus, we have not analysed the effects of the drug and genetic manipulation on other parts of the mutant brain.

13. The title of the article, abstract and other places within the article overstate the relevance of their findings. Blocking the activation of the NLRP3 inflammasome has not been demonstrated to be a general strategy for ataxias and needs to be more specific and narrowly related to the *Elp6*/*wobbly* mice, unless the authors test their findings in multiple ataxia models. The title and statements like “defines a novel therapeutic approach for a severe class of disorders where currently none exist” and “NLRP3 pathway is a putative target for the treatment of ataxias of diverse aetiologies” should be deleted. This would be okay to discuss as a future experiment in the discussion but there is NO data presented to support this general conclusion in this manuscript.

33. We have revised the title and the manuscript in this regard.

14. Figure 7 is highly speculative and should not be presented as the mechanism underlying neurodegeneration in the *wobbly* mouse.

34. We initially added this figure to provide a simplified summary and illustration of our findings for the non-expert reader. We have removed this figure from the manuscript according to the reviewer's suggestion.
15. Image shown in Fig. 7b does not appear to be representative compared to the data shown in Fig. 7b.
35. We have updated this figure (please refer to Fig. 7b in the revised version of the manuscript).
16. The number of animals used in most of the behavioral, histological and electrophysiological tests is missing and should be indicated.
36. We have indicated the number of animals used for all experiments in figure legends.
17. Additional minor concerns:
- Why is there an asterisk for a p value of 0.49 in Supplemental Table 1 – is this a typo?
37. This was a typographical error since the value of AP threshold is 0.049 mV (and not 0.49 as previously stated). This is corrected now in Supplementary Table 1.
- Panels in most of the figures are way too small. The panels and the fonts use for labels needs to be much larger.
38. This has been revised, however, overall this is a function of this particular Nature Communications format. The figures themselves whilst physically small in this format are high resolution so magnify well. The editor will be provided with highest quality images as separate files in the event of acceptance and of course we are happy to provide them to this reviewer ahead of this time.
- Describe what the red bars in Figure 5 indicate. There is no explanation in the text or the legend.
39. This figure is removed. Please refer to the response to comment 5 in this regard.

Reviewer #4 (Remarks to the Author):

In this article Kojic et al evaluate the effects of NLRP3 blockade in a model of neuronal degeneration. The characterisation of the wobbly mouse is sound and nicely pinpoints specific gene targets. However, the rationale for targeting microglia (or NLRP3) and the actual analysis is rather weak and lacks a thorough study in order to support the authors claims. My main comments can be summarised as below:

Main points

1. As the authors nicely introduce, cerebellar ataxias are a complex group of diseases and, although microgliosis seems associated with the progression of the disease, it seems that the

evidence for this being a driving force of it is weak. It could just be a consequence of the over and progressive neurodegeneration. In this sense, what is the evidence for microglia being causative in these diseases? Or, alternatively, how is the current model representative of the human diseases so we can extrapolate findings about microglia? As it stands, it is difficult to gauge the impact of the current findings based on a novel model not yet explored much.

40. As discussed in other responses, our data indicate that NLRP3 activation contributes significantly to the rate of neurodegeneration in wobbly mice but the driving “insult” is likely proteotoxic stress in PNs due to subtle mutation of the Elongator complex. Gliosis is present in ataxic patients (for references refer to the manuscript), however, it has not been characterised in detail so far and data are sparse on human patients when it comes to the relationship between gliosis, neuroinflammation and disease progression. Of course in the long term one way to define this relationship may be to address this empirically through clinical trial of NLRP3 inhibitors.

2. The characterisation of the model is excellent, but being this a novel mouse model, the shown characterisation seems too restricted to the cerebellum, and a more comprehensive characterisation is needed (or needs to be show more than just commented in the text). For example, is survival affected? Is there a different phenotype in males vs females? Do any other areas undergo some degree of neurodegeneration or gliosis (in particular hippocampus because of similarities of cell populations)?

41. Survival is not affected in mutant mice as stated on page 5 in the revised manuscript and showed in Fig. 2b. We have provided data showing no difference in the phenotype of male and female animals (please refer to Supplementary Fig. 1 and Supplementary Fig. 2). In regards to other parts of the brain being affected, please refer to the response to comment 2. The hippocampus is not affected in *wobbly* mice as shown in Supplementary Fig. 3 in the revised manuscript.

3. The characterisation of the gliotic response in the cerebellum of wobbly mice is weak. Only a few micrographs are show, lacking proper analysis. In order to understand the correlation of the neurodegeneration with gliosis, it is necessary to perform a time-course quantification of the number of microglia and astrocytes, as well as the expression of the selected inflammatory markers. Moreover, magnified confocal images are needed in order to better resolve the presence of ASC in glial cells.

42. Following the recommendation, time-course quantification of microgliosis, ASC nucleation and astrogliosis is now shown in Fig. 6 and Supplementary Fig. 9 in the revised manuscript. Magnified confocal images showing the presence of ASC specks in microglial cells are now included in Fig. 6.

4. The authors use a drug MCC950 to block NLRP3, but little information is provided about the action of this compound. What is the precise mode of action? Have you got any data demonstrating this drug is brain penetrant? At what dose?

43. For the mode of action of MCC950, please refer to Coll et al, 2015 (the reference is provided on page 14 of the manuscript). Regarding the brain penetrance, the MCC950 dose used in this study was established in a pilot study where we assessed the penetrance of the drug into the brain tissue of mice at levels above the IC₅₀ of the drug (this is detailed on page 34 and 35 in the revised version of the manuscript).

Statement around dosing - “Mice were dosed orally via drinking water (0.3mg/ml) *ad libitum* starting at P21 until sacrificed at P120.”, as detailed in the revised manuscript on page 34. The complete study on the brain penetrance of MCC950 constitutes a part of another paper performed by Woodruff et al that is currently under revision in *Science*. Data available upon acceptance of that manuscript. Given the molecular and phenotypic response of the *wobbly* cerebellum we observe there is no doubt that that MCC950 is present in the CNS at an active concentration using the dose and regime we describe.

5. The approaches (pharmacological and genetic) utilised to block NLRP3 seem to cause a transient and modest amelioration. Why is this a transient on behaviour? Moreover, the analysis generally lacks controls, and WT and WT+drug (or genetic controls) should be included, in order to appreciate the magnitude of the effects on PN survival, and also be able to perform adequate statistical comparisons. Currently, comparing wobbly vs treated is incomplete and would provide a biased view of the real effect size.

44. The effect of blocking the NLRP3 inflammasome on the ataxic phenotype is significant at P60-P80, and not beyond this age. The neuroprotective effect of the NLRP3 inhibition reflects on the behavior, i.e. it results in a delay of the onset of the ataxic phenotype. We now provide data on behavioral analyses of MCC950-treated wild-type, NLRP3 KO and Casp-1 KO animals in the revised version of the manuscript (please refer to Fig. 7 and Fig. 8).

Minor points:

1. Staining microglia for Iba1 has little meaning, beyond understanding changes in number. It would be highly recommended to stain for activation markers such as MHCII or IL1b, in order to better understand this response to injury.

45. Using Iba-1 marker, we showed the active microgliosis in *wobbly* brains based on changes in the number and morphology of microglia. For activation markers, we chose ASC immuno-staining since ASC speck formation is used as a robust readout for inflammasome activation (Stutz et al, 2013), which is the focus of this study.

2. It would be very useful to provide additional behavioural analysis. Being a model of ataxia, it would be expected to observe early changes in rotarod, climbing test or grip tests, and this would reinforce the validity of the model considerably.

46. We have provided these data in the revised version of the manuscript in Supplementary Fig. 2. The additional behavioural tests include rotarod, balance beam and Catwalk system.

REFERENCES

- Alings, F. et al, An evolutionary approach uncovers a diverse response of tRNA 2-thiolation to elevated temperatures in yeast. *RNA* 21, 202-212, (2015).
- Coll, R. C. et al. A small molecule inhibitor of the NLRP3 inflammasome is a potential therapeutic for inflammatory diseases. *Nature Medicine*. 21; 248-255. (2015).
- Gustin, A. et al. NLRP3 inflammasome is expressed and functional in mouse brain microglia but not in astrocytes. *PloS one* 10, e0130624 (2015).
- Hetz C and Mollereau B. Disturbance of endoplasmic reticulum proteostasis in neurodegenerative diseases. *Nat. Rev. Neurosci*. 15; 233-49. (2014)
- Karlsborn T, et al. Familial dysautonomia (FD) patients have reduced levels of the modified wobble nucleoside mcm(5)s(2)U in tRNA. *Biochem. Biophys. Res. Commun*. 454; 441-5. (2014).
- Lee, J. W. et al. Editing-defective tRNA synthetase causes protein misfolding and neurodegeneration. *Nature*. 443; 50 -5. (2006).
- Nedialkova, D.D. and Leidel, S.A. Optimization of Codon Translation Rates via tRNA Modifications Maintains Proteome Integrity. *Cell*. 161;1606-1618. (2015).
- Sharma, K. et al. Cell type– and brain region–resolved mouse brain proteome. *Nature Neuroscience*. 18; 1819-31. (2015)
- Song, L. et al. NLRP3 Inflammasome in Neurological Diseases, from Functions to Therapies. *Frontiers in Cellular Neuroscience*. 11; 63 - (2017).
- Stutz, A., et al. ASC speck formation as a readout for inflammasome activation. *The Inflammasome: Methods and Protocols*, 91-101 (2013).
- Tomomura, M. et al. Purification of Purkinje cells by fluorescence - activated cell sorting from transgenic mice that express green fluorescent protein. *European Journal of Neuroscience*. 14; 57-63. (2001)
- Williams RW, Herrup K. The control of neuron number. *Annu Rev Neurosci*. 11:423-53.

Appendix 1.

Summary of additional data.

Analysed other CNS and PNS structures in *wobbly* mice:

- H&E staining of the whole brain; magnified images of the hippocampus showing no differences between *wobbly* and control mice (no neurodegeneration, no gliosis);
- H&E staining of the spinal cord; magnified images of the ventral and dorsal horn showing no differences between *wobbly* and control mice (no neurodegeneration, no gliosis);
- Other organs analysed by necroscopy in *wobbly* mice – the list included in the manuscript.

Behavioural testing:

- Individual scores/graphs are provided for the ledge test, hindlimb clasping and locomotor activity (for each test wild-type, heterozygote and homozygote *wobbly* mice are analysed);
- Individual graphs for ledge test, hindlimb clasping and locomotor activity – male vs. female *wobbly* mice tested;
- Composite phenotype graph – male vs. female *wobbly* mice tested;
- Additional behavioural tests: Rotarod, Catwalk and Balance beam test (for each test wild-type, heterozygote and homozygote *wobbly* animals analysed);
- Rotarod, Catwalk and Balance beam test – male vs. female *wobbly* mice tested;
- Composite phenotype graph – Pcp2-Elp6-HA mice tested (we showed that the complementation experiment rescues the motor deficit in *wobbly* mice).

We provided graphs showing survival in *wobbly* (unaffected) and Elp6L126Q/KO mice (affected; mice do not survive beyond P60).

Time-course quantification of gliosis in *wobbly* mice:

- Immuno-fluorescence of *wobbly* cerebellar sections at P21, P60 and P120 using anti: Iba-1, GFAP and ASC antibodies, followed by quantification of micro- and astro-gliosis and ASC specks in the field of view;
- Magnified (x63) confocal images showing ASC nucleation within microglia in *wobbly* mice.

MCC950-treatment of *wobbly* mice graph:

- Included data on wild-type and MCC950-treated wild-type as control animals in the graph.

NLRP3 and Casp1 KO *wobbly* mice graph:

- Included data on NLRP3 and Casp1 KO control animals in the graph.

Quantification of microgliosis and ASC nucleation in NLRP3 KO and Casp1 KO *wobbly* mice:

- Immuno-fluorescence using anti: Iba-1 and ASC antibodies, followed by quantification of activated microglia and ASC specks in the field of view.

tRNA modification data:

- Analyses of additional tRNA molecule (tRNA^{Cys}) which showed a very similar difference in binding capacity of mutated Elp456 by Elongator. Comparison of two tRNA types, a modifiable (tRNA^{Ala}) and non-modifiable (tRNA^{Cys}) by Elongator, showed that the Elongator complex in *wobbly* mice has a decreased capacity to bind tRNA molecules.
- NanoDSF experiments that show an increase in the amount of aggregated Elp456 protein in the L126Q samples.

REVIEWERS' COMMENTS:

Reviewer #2 (Remarks to the Author):

The authors answered all my concerns.

Reviewer #3 (Remarks to the Author):

This is a very interesting manuscript that describes a novel mutation in the elongator complex that results in Purkinje cell degeneration in ataxic mice. The authors have addressed previous concerns and have added substantial new data in support of their conclusions that a point mutation in the gene encoding Elongator complex subunit 6 (Elp6), destabilizes the complex, compromises its function which results in translational dysregulation, protein misfolding, proteotoxic stress and eventual neuronal death. My only remaining suggestion is to add a couple of online images to support the statements that inflammation is limited to the cerebellum.

Reviewer #4 (Remarks to the Author):

The authors have addressed the raised comments very nicely, and the manuscript has been significantly improved to provide compelling evidence supporting the conclusions.